# Robust Fusion Kalman Estimator of the Multi-Sensor Descriptor System with Multiple Types of Noises and Packet Loss

**DOI:** 10.3390/s23156968

**Published:** 2023-08-05

**Authors:** Jie Zheng, Wenxia Cui, Sian Sun

**Affiliations:** School of Mathematics, Physics and Statistics, Shanghai University of Engineering Science, Shanghai 201620, China; cuiwx423@163.com (W.C.); sunsian@163.com (S.S.)

**Keywords:** descriptor system, Kalman estimator, unified measurement model, multi-sensor, multiplicative noises, uncertain-variance noises

## Abstract

Under the influence of multiple types of noises, missing measurement, one-step measurement delay and packet loss, the robust Kalman estimation problem is studied for the multi-sensor descriptor system (MSDS) in this paper. Moreover, the established MSDS model describes uncertain-variance noises, multiplicative noises, time delay and packet loss phenomena. Different types of noises and packet loss make it more difficult to build the estimators of MSDS. Firstly, MSDS is transformed to the new system model by applying the singular value decomposition (SVD) method, augmented state and fictitious noise approach. Furthermore, the robust Kalman estimator is constructed for the newly deduced augmented system based on the min-max robust estimation principle and Kalman filter theory. In addition, the given estimator consists of four parts, which are the usual Kalman filter, predictor, smoother and white noise deconvolution estimator. Then, the robust fusion Kalman estimator is obtained for MSDS according to the relation of augmented state and the original system state. Simultaneously, the robustness is demonstrated for the actual Kalman estimator of MSDS by using the mathematical induction method and Lyapunov’s equation. Furthermore, the error variance of the obtained Kalman estimator is guaranteed to the upper bound for all admissible uncertain noise variance. Finally, the simulation example of a circuit system is examined to illustrate the performance and effectiveness of the robust estimators.

## 1. Introduction

The descriptor system is also a singular system, which has a broader structure than the normal system. Furthermore, the descriptor system can describe the non-causal phenomena in real systems, such as robot systems, power systems, image modeling, and economic systems [1,2,3]. The state estimation problem of the descriptor system has been a popular topic in recent years. Many research results and methods have been obtained to solve the estimation problem [4,5,6,7,8,9,10,11,12]. Based on the reduced-order Kalman estimation algorithm [13,14], the singular value decomposition (SVD) method for the descriptor system is presented in [4,7]. The authors of [5] give the least squares method and the maximum likelihood method for the descriptor systems, respectively. In [8], the time domain Wiener filter for the descriptor system is proposed by using the modern time series analysis method. However, the above estimation problems are only studied for the known general descriptor systems.

Moreover, it is well known that the estimator based on the classical Kalman filtering requires that noise statistics and the model parameters are exactly known [11]. However, in many practical systems, there exist many uncertainties such as modelling errors, unmodeled dynamic, random perturbations, missing measurements, measurement delays, multiplicative noises and so on [15,16,17,18]. In order to solve the effect of the uncertainty, the robust estimation is studied for an uncertain system [11]. At present, for the uncertain descriptor system, the Kalman robust filter and predictor are presented [12]. The robust time-varying estimator is proposed for descriptor systems with random one-step measurement delay by using the SVD method, the augmented method, and the fictitious noise approach [19]. However, it should be noted that reference [19] only considers the descriptor with a one-step measurement delay, and other uncertainties are not considered. In [20], the robust centralized and weighted observation fusion (CAWOF) prediction algorithm is derived for the uncertain MSDS with multiplicative noise by using the SVD method and the minimax robustness estimation criterion. Reference [20] only considers the descriptor system with multiplicative noise and uncertain noise. However, packet loss and measurement delay problems have not been taken into account. In [21], the uncertain-variance noises and packet loss problems are solved in the MSDS; however, the effects of multiplicative noise and measurement delay are not considered in the MSDS.

In addition, the estimation accuracy and performance of a single sensor descriptor system can be easily affected by the stability and reliability of the sensor [22]. To improve estimation accuracy and guarantee performance of the considered system, a multi-sensor system has been widely used [23]. For the multi-sensor descriptor system, Kalman filtering is a fundamental tool due to its recursive structure and excellent performance. In general, the fusion method of the Kalman filter can be categorized into three types: centralized fusion, measurement fusion, and distributed state fusion method [24,25]. In [24,26], the authors present distributed fusion algorithms that use optimally weighted fusion criteria with a matrix weight, a diagonal matrix weight, and a scalar weight. These algorithms the address estimation problems in multi-sensor systems, which are typically studied based on the known parameters of the system model and the complete known noise statistical structure. In [25], the fusion Kalman filter algorithm deals with an uncertain nonsingular system with multiplicative noises, missing measurements, and linearly correlated white noises with uncertain variances. However, for a multi-sensor networked descriptor control system, the distributed fusion robust Kalman filter algorithm is proposed in [27]. However, reference [27] only considers uncertain-variance correlated noises and missing measurement problems of the multi-sensor networked descriptor control system.

To date, the robust fusion estimation problem is not solved for MSDS with uncertain-variance noises, multiplicative noises and a unified measurement model, which totally include five kinds of uncertainties which are uncertain-variance noises, multiplicative noises, missing measurements, one-step measurement delays and packet dropouts. Motivated by the aforementioned analysis, for MSDS with the above five uncertainties, the robust estimation problem will be studied. The main contributions and innovations of this paper are as follows: (1) The considered MSDS is novel and challenging, which includes uncertain-variance noises, multiplicative noises, missing measurements, one-step measurement delays and packet dropouts. (2) Applying the SVD method, the augmented state method and the fictitious white noises method, MSDS is transformed to a new standard system only with uncertain-variance noise. (3) Based on the Kalman filter and the relations of the original MSDS and the newly obtained system, the robust Kalman estimators are given for MSDS and the newly obtained augmented system. (4) The robustness is proved for the proposed estimators by using the Lyapunov equation approach and the mathematical induction method.

This paper is organized into seven sections. In Section 2, the system model is given. In Section 3, a new standard augmented state model is presented. The robust Kalman estimator for descriptor system is discussed in Section 4. In Section 5, a robust analysis is discussed. Section 6 presents the numerical simulation results. Finally, Section 7 provides the conclusion.

## 2. System Description and Preliminaries

Consider MSDS with uncertain-variance noises, multiplicative noises and a unified measurement model
(1)$$\begin{array}{cc} \; & Mx(t+1)=\Phi x(t)+\Gamma \omega (t)+Bu(t), \end{array}$$
(2)$$\begin{array}{cc} \; & z_{0i}\left(t\right)=\left(H_{i}+\sum_{l=1}^{n_{a}}a_{il}\left(t\right)H_{il}\right)x\left(t\right), \end{array}$$
(3)$$\begin{array}{cc} \; & z_{i}\left(t\right)=\gamma_{i}\left(t\right)z_{0i}\left(t\right)+\nu_{i}\left(t\right), \end{array}$$
(4)$$\begin{array}{cc} \; & y_{i}\left(t\right)=\alpha_{i}\left(t\right)z_{i}\left(t\right)+\left(1-\alpha_{i}\left(t\right)\right)\beta_{i}\left(t\right)z_{i}\left(t-1\right)+\left(1-\alpha_{i}\left(t\right)\right)\left(1-\beta_{i}\left(t\right)\right)y_{i}\left(t-1\right),\;i=0,1,\cdots ,L \end{array}$$
where *t* is a discrete time, $x\left(t\right)\in R^{n}$ is the state, u(t) is the input, $\omega \left(t\right)\in R^{n_{w}}$ is additive process noise, $\nu_{i}\left(t\right)\in R^{m_{i}}$ is additive measurement noise, $z_{0i}\left(t\right)\in R^{m_{i}}$ is the *i*th noise-free measurement, $a_{il}\left(t\right)\in R^{1}$ is multiplicative state-dependent noise, $z_{i}\left(t\right)\in R^{m_{i}}$ is the measurement of the *i*th sensor, $y_{i}\left(t\right)\in R^{m_{i}}$ is the measurement received by estimator to be designed, $n_{a}$ and *L* are the number of multiplicative noises and sensors, respectively. *M*, Φ, Γ, *B* and $H_{i}$ are constant matrices with suitable dimensions.

**Assumption 1.** 
*M is a singular matrix, $rank\left(M\right)=n_{1}$, $n_{1}<n$, that is, det M=0, and the system (Equation 1) is regular.*


**Assumption 2.** 
*$\alpha_{i}\left(t\right)$, $\beta_{i}\left(t\right)$ and $\gamma_{i}\left(t\right)$(i=0,1,⋯,L) are mutually independent random sequences, obeying Bernoulli distributions with known probabilities of taking *1* or *0*, such that*

(5)
$$\begin{array}{cc} \; & \mathrm{Prob}\left\{\alpha_{i}\left(t\right)=1\right\}=\lambda_{\alpha_{i}},\;\;\;\mathrm{Prob}\left\{\alpha_{i}\left(t\right)=0\right\}=1-\lambda_{\alpha_{i}},\;\;\;0\leq \lambda_{\alpha_{i}}\leq 1, \end{array}$$


(6)
$$\begin{array}{cc} \; & \mathrm{Prob}\left\{\beta_{i}\left(t\right)=1\right\}=\lambda_{\beta_{i}},\;\;\;\mathrm{Prob}\left\{\beta_{i}\left(t\right)=0\right\}=1-\lambda_{\beta_{i}},\;\;\;0\leq \lambda_{\beta_{i}}\leq 1, \end{array}$$


(7)
$$\begin{array}{cc} \; & \mathrm{Prob}\left\{\gamma_{i}\left(t\right)=1\right\}=\lambda_{\gamma_{i}},\;\;\;\mathrm{Prob}\left\{\gamma_{i}\left(t\right)=0\right\}=1-\lambda_{\gamma_{i}},\;\;\;0\leq \lambda_{\gamma_{i}}\leq 1,\;\;\;i=0,1,\cdots ,L, \end{array}$$

*from Assumption 2, it follow that*

(8)
$$\begin{array}{c} E\left[\alpha_{i}\left(t\right)\right]=E\left[\alpha_{i}^{2}\left(t\right)\right]=\lambda_{\alpha i},\;\;\;E\left[\beta_{i}\left(t\right)\right]=E\left[\beta_{i}^{2}\left(t\right)\right]=\lambda_{\beta i},\;\;\;E\left[\gamma_{i}\left(t\right)\right]=E\left[\gamma_{i}^{2}\left(t\right)\right]=\lambda_{\gamma i}, \end{array}$$

*zero-means white noises $\alpha_{0i}\left(t\right)$, $\beta_{0i}\left(t\right)$ and $\gamma_{0i}\left(t\right)$ are defined as follows:*

(9)
$$\begin{array}{c} \alpha_{0i}\left(t\right)=\alpha_{i}\left(t\right)-\lambda_{\alpha i},\;\;\beta_{0i}\left(t\right)=\beta_{i}\left(t\right)-\lambda_{\beta i},\;\;\gamma_{0i}\left(t\right)=\gamma_{i}\left(t\right)-\lambda_{\gamma i}, \end{array}$$

*it follow that*

(10)
$$\begin{array}{cc} \; & E\left[\alpha_{0i}\left(t\right)\right]=0,\;\;E\left[\alpha_{0i}^{2}\left(t\right)\right]=\lambda_{\alpha i}\left(1-\lambda_{\alpha i}\right)≜\lambda_{\alpha 0i},\;\;E\left[\alpha_{0i}\left(t\right)\alpha_{0j}\left(k\right)\right]=0,\;\;\;i\neq j,\;\;\;\forall t,k, \end{array}$$


(11)
$$\begin{array}{cc} \; & E\left[\beta_{0i}\left(t\right)\right]=0,\;\;E\left[\beta_{0i}^{2}\left(t\right)\right]=\lambda_{\beta i}\left(1-\lambda_{\beta i}\right)≜\lambda_{\beta 0i},\;\;\;E\left[\beta_{0i}\left(t\right)\beta_{0j}\left(k\right)\right]=0,\;\;\;i\neq j,\;\;\;\forall t,k, \end{array}$$


(12)
$$\begin{array}{cc} \; & E\left[\gamma_{0i}\left(t\right)\right]=0,\;\;\;E\left[\gamma_{0i}^{2}\left(t\right)\right]=\lambda_{\gamma i}\left(1-\lambda_{\gamma i}\right)≜\lambda_{\gamma 0i},\;\;\;E\left[\gamma_{0i}\left(t\right)\gamma_{0j}\left(k\right)\right]=0,\;\;\;i\neq j,\;\;\;\forall t,k. \end{array}$$



**Assumption 3.** 
*ω(t), $\nu_{i}\left(t\right)$ and $a_{il}\left(t\right)$ are mutually independent white noises with zero means and the unknown actual variance are ${\bar{Q}}_{w}$, ${\bar{R}}_{i}$ and ${\bar{\sigma }}_{\alpha il}$, respectively, and*

(13)
$$\begin{array}{c} E\left[\omega \left(t\right)\omega^{T}\left(t\right)\right]={\bar{Q}}_{w},\;\;\;E\left[\nu_{i}\left(t\right)\nu_{j}^{T}\left(t\right)\right]={\bar{R}}_{i}\delta_{ij},\;\;E\left[a_{il}\left(t\right)a_{jl}^{T}\left(t\right)\right]={\bar{\sigma }}_{\alpha il}\delta_{ij}. \end{array}$$



The unknown actual variance are, respectively, have known conservative upper bounds, which are
(14)$$\begin{array}{c} {\bar{Q}}_{w}\leq Q_{w},\;\;\;{\bar{R}}_{i}\leq R_{i},\;\;{\bar{\sigma }}_{\alpha il}\leq \sigma_{\alpha il}. \end{array}$$

**Remark 1.** 
*In real-world measurement, time delay and packet loss may occur at any time. The measurement models (2)–(4) describe a unified measurement model by introducing random sequences $\alpha_{i}\left(t\right)$, $\beta_{i}\left(t\right)$ and $\gamma_{i}\left(t\right)$, which include the missing measurements, one-step delay measurement and packet dropouts. If $\gamma_{i}\left(t\right)=1$, $\alpha_{i}\left(t\right)=1$, then $y_{i}\left(t\right)=z_{i}\left(t\right)$. If $\gamma_{i}\left(t\right)=0$, $\alpha_{i}\left(t\right)=1$, then $y_{i}\left(t\right)=\nu_{i}\left(t\right)$, which means measurement missed. If $\alpha_{i}\left(t\right)=0$, $\beta_{i}\left(t\right)=1$, then $y_{i}\left(t\right)=z_{i}\left(t-1\right)$, which means that there is one-step measurement delay. If $\alpha_{i}\left(t\right)=0$, $\beta_{i}\left(t\right)=0$, then $y_{i}\left(t\right)=y_{i}\left(t-1\right)$, which means packet dropout.*


## 3. New Standard Augmented State Model with Uncertain-Variance Fictitious Noises

Applying the SVD approach, there are non-singular matrices *P* and *Q* satisfying
(15)$$\begin{array}{cc} \; & PMQ=\left[\begin{array}{cc} M_{1} & 0\\ 0 & 0 \end{array}\right],\;\;M_{1}=\mathrm{diag}\left\{\sigma_{1},\cdots ,\sigma_{n_{1}}\right\}, \end{array}$$
(16)$$\begin{array}{cc} \; & P\Phi Q=\left[\begin{array}{cc} \Phi_{11} & \Phi_{12}\\ \Phi_{21} & \Phi_{22} \end{array}\right],\;P\Gamma =\left[\begin{array}{c} \Gamma_{1}\\ \Gamma_{2} \end{array}\right],\;\;PB=\left[\begin{array}{c} B_{1}\\ B_{2} \end{array}\right],\;\;H_{i}Q=\left[H_{i1},\;\;H_{i2}\right], \end{array}$$
letting
(17)$$\begin{array}{c} x\left(t\right)=Q\left[\begin{array}{c} x_{1}\left(t\right)\\ x_{2}\left(t\right) \end{array}\right], \end{array}$$
substituting (Equation 15) and (Equation 16) into (Equation 1) yields
(18)$$\begin{array}{c} \left[\begin{array}{cc} M_{1} & 0\\ 0 & 0 \end{array}\right]\left[\begin{array}{c} x_{1}\left(t+1\right)\\ x_{2}\left(t+1\right) \end{array}\right]=\left[\begin{array}{cc} \Phi_{11} & \Phi_{12}\\ \Phi_{21} & \Phi_{22} \end{array}\right]\left[\begin{array}{c} x_{1}\left(t\right)\\ x_{2}\left(t\right) \end{array}\right]+\left[\begin{array}{c} \Gamma_{1}\\ \Gamma_{2} \end{array}\right]\omega \left(t\right)+\left[\begin{array}{c} B_{1}\\ B_{2} \end{array}\right]u\left(t\right), \end{array}$$
then we have two new subsystems
(19)$$\begin{array}{cc} x_{1}\left(t+1\right) & =J_{x_{1}}x_{1}\left(t\right)+U_{x_{1}}\omega \left(t\right)+G_{x_{1}}u\left(t\right), \end{array}$$
(20)$$\begin{array}{cc} x_{2}\left(t\right) & =J_{x_{2}}x_{1}\left(t\right)+U_{x_{2}}\omega \left(t\right)+G_{x_{2}}u\left(t\right), \end{array}$$
where $J_{x_{1}}=M_{1}^{-1}\left(\Phi_{11}-\Phi_{12}\Phi_{22}^{-1}\Phi_{21}\right)$, $U_{x_{1}}=M_{1}^{-1}\left(\Gamma_{1}-\Phi_{12}\Phi_{22}^{-1}\Gamma_{2}\right)$, $G_{x_{1}}=M_{1}^{-1}\left(B_{1}-\Phi_{12}\Phi_{22}^{-1}B_{2}\right)$, $J_{x_{2}}=-\Phi_{22}^{-1}\Phi_{21}$, $U_{x_{2}}=-\Phi_{22}^{-1}\Gamma_{2}$, $G_{x_{1}}=-\Phi_{22}^{-1}B_{2}$, $H_{i}Q$ in (Equation 16) and (Equation 17) are substituted into (2), then it is easy to obtain
$$\begin{array}{cc} z_{0i}\left(t\right) & =H_{i1}x_{1}\left(t\right)+H_{i2}x_{2}\left(t\right)+\left(\sum_{l=1}^{n_{a}}\alpha_{il}\left(t\right)H_{il}\right)x\left(t\right), \end{array}$$
substituting (20) into $z_{0i}\left(t\right)$ yields
(21)$$\begin{array}{c} z_{0i}\left(t\right)=\left(H_{i1}+H_{i2}J_{x_{2}}\right)x_{1}\left(t\right)+H_{i2}U_{x_{2}}\omega \left(t\right)+H_{i2}G_{x_{2}}u\left(t\right)+\left(\sum_{l=1}^{n_{a}}\alpha_{il}\left(t\right)H_{il}\right)x\left(t\right), \end{array}$$
substituting (21) into (3), it is easy to obtain
(22)$$\begin{array}{cc} z_{i}\left(t\right)= & \gamma_{i}\left(t\right)\left(H_{i1}+H_{i2}J_{x_{2}}\right)x_{1}\left(t\right)+\gamma_{i}\left(t\right)H_{i2}U_{x_{2}}\omega \left(t\right)+\gamma_{i}\left(t\right)H_{i2}G_{x_{2}}u\left(t\right)\\ \; & +\gamma_{i}\left(t\right)\left(\sum_{l=1}^{n_{a}}\alpha_{il}\left(t\right)H_{il}\right)x\left(t\right)+\nu_{i}\left(t\right), \end{array}$$
from (Equation 9), we have $\gamma_{i}\left(t\right)=\gamma_{0i}\left(t\right)+\lambda_{\gamma i}$, in (Equation 22), replacing $\gamma_{i}\left(t\right)$ by $\gamma_{0i}\left(t\right)+\lambda_{\gamma i}$ yields
(23)$$\begin{array}{cc} z_{i}\left(t\right) & =\lambda_{\gamma i}\left(H_{i1}+H_{i2}J_{x_{2}}\right)x_{1}\left(t\right)+\left(\gamma_{0i}\left(t\right)+\lambda_{\gamma i}\right)H_{i2}G_{x_{2}}u\left(t\right)+\nu_{zi}\left(t\right), \end{array}$$
where $\nu_{zi}\left(t\right)=\gamma_{0i}\left(t\right)\left(H_{i1}+H_{i2}J_{x_{2}}\right)x_{1}\left(t\right)+\left(\gamma_{0i}\left(t\right)+\lambda_{\gamma i}\right)H_{i2}U_{x_{2}}\omega \left(t\right)+\left(\gamma_{0i}\left(t\right)+\lambda_{\gamma i}\right)$ × $\left(\sum_{l=1}^{n_{a}}\alpha_{il}\left(t\right)H_{il}\right)x\left(t\right)+\nu_{i}\left(t\right)$, substituting (Equation 23) into (4), replacing $\alpha_{i}\left(t\right)$ by $\alpha_{0i}\left(t\right)+\lambda_{\alpha i}$ and replacing $\beta_{i}\left(t\right)$ by $\beta_{0i}\left(t\right)+\lambda_{\beta i}$ yield
(24)$$\begin{array}{cc} y_{i}\left(t\right)= & \lambda_{\alpha i}\lambda_{\gamma i}\left(H_{i1}+H_{i2}J_{x_{2}}\right)x_{1}\left(t\right)+\left(\alpha_{0i}\left(t\right)+\lambda_{\alpha i}\right)\left(\gamma_{0i}\left(t\right)+\lambda_{\gamma i}\right)H_{i2}G_{x_{2}}u\left(t\right)\\ \; & +\left(1-\lambda_{\alpha i}\right)\lambda_{\beta i}z_{i}\left(t-1\right)+\left(1-\lambda_{\alpha i}\right)\left(1-\lambda_{\beta i}\right)y_{i}\left(t-1\right)+\nu_{yi}\left(t\right), \end{array}$$
where $\nu_{yi}\left(t\right)=\left(\alpha_{0i}\left(t\right)+\lambda_{\alpha i}\right)\left(\gamma_{0i}\left(t\right)+\lambda_{\gamma i}\right)\left(\sum_{l=1}^{n_{a}}\alpha_{il}\left(t\right)H_{il}\right)x\left(t\right)+\left(\alpha_{0i}\left(t\right)+\lambda_{\alpha i}\right)\left(\gamma_{0i}\left(t\right)+\lambda_{\gamma i}\right)H_{i2}U_{x_{2}}\omega \left(t\right)+\left(\alpha_{0i}\left(t\right)+\lambda_{\alpha i}\right)\nu_{i}\left(t\right)+\lambda_{\alpha i}\gamma_{0i}\left(t\right)\left(H_{i1}+H_{i2}J_{x_{2}}\right)x_{1}\left(t\right)+\alpha_{0i}\left(t\right)\gamma_{0i}\left(t\right)\left(H_{i1}+H_{i2}J_{x_{2}}\right)x_{1}\left(t\right)+\lambda_{\gamma i}\alpha_{0i}\left(t\right)\left(H_{i1}+H_{i2}J_{x_{2}}\right)x_{1}\left(t\right)+\left(1-\lambda_{\alpha i}\right)\beta_{0i}\left(t\right)z_{i}\left(t-1\right)-\lambda_{\beta i}\alpha_{0i}\left(t\right)z_{i}\left(t-1\right)-\alpha_{0i}\left(t\right)\beta_{0i}\left(t\right)z_{i}\left(t-1\right)-\left(1-\lambda_{\alpha i}\right)\beta_{0i}\left(t\right)y_{i}\left(t-1\right)-\left(1-\lambda_{\beta i}\right)\alpha_{0i}\left(t\right)y_{i}\left(t-1\right)+\alpha_{0i}\left(t\right)\beta_{0i}\left(t\right)y_{i}\left(t-1\right)$. In order to facilitate the calculation, it is necessary to simplify $\nu_{yi}\left(t\right)$. New parameters $C_{ui}\left(t\right)$ and $H_{ui}\left(t\right)$(u=1,2,3,4) are defined, then we can rewrite $\nu_{yi}\left(t\right)$ as
(25)$$\begin{array}{cc} \nu_{yi}\left(t\right)= & \left(\alpha_{0i}\left(t\right)+\lambda_{\alpha i}\right)\left(\gamma_{0i}\left(t\right)+\lambda_{\gamma i}\right)\left(\sum_{l=1}^{n_{a}}\alpha_{il}\left(t\right)H_{il}\right)x\left(t\right)+\left(\alpha_{0i}\left(t\right)+\lambda_{\alpha i}\right)\nu_{i}\left(t\right)\\ \; & +\left(\alpha_{0i}\left(t\right)+\lambda_{\alpha i}\right)\left(\gamma_{0i}\left(t\right)+\lambda_{\gamma i}\right)H_{i2}U_{x_{2}}\omega \left(t\right)+\lambda_{\alpha i}\gamma_{0i}\left(t\right)\left(H_{i1}+H_{i2}J_{x_{2}}\right)x_{1}\left(t\right)\\ \; & +\sum_{u=1}^{4}C_{ui}\left(t\right)H_{cui}\left[\begin{array}{c} x_{1}\left(t\right)\\ z_{i}\left(t-1\right)\\ y_{i}\left(t-1\right) \end{array}\right], \end{array}$$
where $C_{1i}=\alpha_{0i}\left(t\right),\;\;C_{2i}=\beta_{0i}\left(t\right),\;\;C_{3i}=\alpha_{0i}\left(t\right)\gamma_{0i}\left(t\right),\;\;C_{4i}=\alpha_{0i}\left(t\right)\beta_{0i}\left(t\right),H_{c1i}=\left[\lambda_{\gamma i}\left(H_{i1}+H_{i2}J_{x_{2}}\right),\;\;-\lambda_{\beta i}I_{mi},\;\;-\left(1-\lambda_{\beta i}\right)I_{mi}\right],H_{c2i}=\left[0,\;\;\left(1-\lambda_{\alpha i}\right)I_{mi},\;\;-\left(1-\lambda_{\alpha i}\right)I_{mi}\right],H_{c3i}=\left[\left(H_{i1}+H_{i2}J_{x_{2}}\right),\;\;0,\;\;0\right],\;\;H_{c4i}=\left[0,\;\;-I_{mi},\;\;I_{mi}\right]$, defining new white noise variances $\sigma_{cui}^{2}=E\left[C_{ui}C_{ui}^{T}\right]$(u=1,2,3,4) as follows
(26)$$\begin{array}{c} \sigma_{c1i}^{2}=\lambda_{\alpha 0i},\;\;\sigma_{c2i}^{2}=\lambda_{\beta 0i},\;\;\sigma_{c3i}^{2}=\lambda_{\alpha 0i}\lambda_{\gamma 0i},\;\;\sigma_{c4i}^{2}=\lambda_{\alpha 0i}\lambda_{\beta 0i}, \end{array}$$
let
(27)$$\begin{array}{c} x_{ai}\left(t\right)=\left[\begin{array}{c} x_{1}\left(t\right)\\ z_{i}\left(t-1\right)\\ y_{i}\left(t-1\right) \end{array}\right],\;\;\omega_{ai}\left(t\right)=\left[\begin{array}{c} \omega_{t}\\ \nu_{zi}\left(t\right)\\ \nu_{yi}\left(t\right) \end{array}\right],\;\;y_{ai}\left(t\right)=y_{i}\left(t\right)-\alpha_{i}\left(t\right)\gamma_{i}\left(t\right)H_{i2}G_{x_{2}}u\left(t\right), \end{array}$$
then it is easy to obtain the new standard augmented state apace model as follows
(28)$$\begin{array}{cc} x_{ai}\left(t+1\right) & =\Phi_{ai}x_{ai}\left(t\right)+\Gamma_{ai}\omega_{ai}\left(t\right)+G_{ai}u\left(t\right), \end{array}$$
(29)$$\begin{array}{cc} y_{ai}\left(t\right) & =H_{ai}x_{ai}\left(t\right)+\nu_{yi}\left(t\right), \end{array}$$
where
(30)$$\begin{array}{cc} \Phi_{ai} & =\left[\begin{array}{ccc} J_{x1} & 0 & 0\\ \lambda_{\gamma i}\left(H_{i1}+H_{i2}J_{x_{2}}\right) & 0 & 0\\ \lambda_{\alpha i}\lambda_{\gamma i}(H_{i1}+H_{i2}J_{x_{2}}) & \left(1-\lambda_{\alpha i}\right)\lambda_{\beta i}I_{mi} & \left(1-\lambda_{\alpha i}\right)\left(1-\lambda_{\beta i}\right)I_{mi} \end{array}\right],\\ \Gamma_{ai} & =\left[\begin{array}{ccc} U_{x1} & 0 & 0\\ 0 & I_{mi} & 0\\ 0 & 0 & I_{mi} \end{array}\right],\;\;G_{ai}=\left[\begin{array}{c} G_{x1}\\ \gamma_{i}\left(t\right)H_{i2}G_{x_{2}}\\ 0 \end{array}\right],\\ H_{ai} & =[\lambda_{\alpha i}\lambda_{\gamma i}\left(H_{i1}+H_{i2}J_{x_{2}}),\;\;\left(1-\lambda_{\alpha i}\right)\lambda_{\beta i}I_{mi},\;\;\left(1-\lambda_{\alpha i}\right)\left(1-\lambda_{\beta i}\right)I_{mi}\right]. \end{array}$$

Non-central second order moments are defined as $X\left(t\right)=E[x\left(t\right)x^{T}\left(t\right)]$, $X_{1}\left(t\right)=E\left[x_{1}\left(t\right)x_{1}^{T}\left(t\right)\right]$ and $X_{ai}\left(t\right)=E\left[x_{ai}\left(t\right)x_{ai}^{T}\left(t\right)\right]$, they satisfy the following Lyapunov equations
(31)$$\begin{array}{cc} \; & X_{1}\left(t+1\right)=J_{x1}X_{1}\left(t\right)J_{x1}^{T}+U_{x1}Q_{w}U_{x1}^{T},\\ \; & X_{ai}\left(t+1\right)=\Phi_{ai}X_{ai}\left(t\right)\Phi_{ai}^{T}+\Gamma_{ai}Q_{wai}\left(t\right)\Gamma_{ai}^{T}, \end{array}$$
and we have corresponding upper values
(32)$$\begin{array}{cc} \; & {\bar{X}}_{1}\left(t+1\right)=J_{x1}{\bar{X}}_{1}\left(t\right)J_{x1}^{T}+U_{x1}{\bar{Q}}_{w}U_{x1}^{T},\\ \; & {\bar{X}}_{ai}\left(t+1\right)=\Phi_{ai}{\bar{X}}_{ai}\left(t\right)\Phi_{ai}^{T}+\Gamma_{ai}{\bar{Q}}_{wai}\left(t\right)\Gamma_{ai}^{T}, \end{array}$$
with initial values $X_{ai}\left(0\right)=\mathrm{diag}\left(P_{01},\;\;0,\;\;0\right),\;\;{\bar{X}}_{ai}\left(0\right)=\mathrm{diag}\left(P_{01},\;\;0,\;\;0\right)$, $P_{0}=\left[\begin{array}{cc} P_{01} & *\\ {}* & * \end{array}\right]$, ${\bar{P}}_{0}=\left[\begin{array}{cc} {\bar{P}}_{01} & *\\ {}* & * \end{array}\right]$.

For the new process noise $\omega_{ai}$ in (Equation 28), it has corresponding conservative variance $Q_{wai}$ and real variance ${\bar{Q}}_{wai}$. Similarly, for new measurement noise $\nu_{yi}\left(t\right)$ in (29), it has corresponding conservative variance $R_{zyi}\left(t\right)$ and real variance ${\bar{R}}_{zyi}\left(t\right)$.

Let $R_{zi}\left(t\right)=E\left[\nu_{zi}\left(t\right)\nu_{zi}^{T}\left(t\right)\right]$, ${\bar{R}}_{zi}\left(t\right)$ is actual variance of $\nu_{zi}\left(t\right)$, the conservative and actual noise variances $R_{zi}\left(t\right)$ and ${\bar{R}}_{zi}\left(t\right)$ are given as follows
(33)$$\begin{array}{cc} R_{zi}\left(t\right)= & \lambda_{\gamma 0i}\left(H_{i1}+H_{i2}J_{x_{2}}\right)X_{1}\left(t\right){\left(H_{i1}+H_{i2}J_{x_{2}}\right)}^{T}+\lambda_{\gamma i}\sum_{l=1}^{n_{a}}\sigma_{\alpha il}^{2}H_{il}X\left(t\right)H_{il}^{T}\\ \; & +\lambda_{\gamma i}H_{i2}U_{x_{2}}Q_{w}{\left(H_{i2}U_{x_{2}}\right)}^{T}+R_{i},\\ {\bar{R}}_{zi}\left(t\right)= & \lambda_{\gamma 0i}\left(H_{i1}+H_{i2}J_{x_{2}}\right){\bar{X}}_{1}\left(t\right){\left(H_{i1}+H_{i2}J_{x_{2}}\right)}^{T}+\lambda_{\gamma i}\sum_{l=1}^{n_{a}}{\bar{\sigma }}_{\alpha il}^{2}H_{il}X\left(t\right)H_{il}^{T}\\ \; & +\lambda_{\gamma i}H_{i2}U_{x_{2}}{\bar{Q}}_{w}{\left(H_{i2}U_{x_{2}}\right)}^{T}+{\bar{R}}_{i}. \end{array}$$

Let $R_{zyi}\left(t\right)=E\left[\nu_{yi}\left(t\right)\nu_{yi}^{T}\left(t\right)\right]$, then ${\bar{R}}_{zyi}\left(t\right)$ is the actual variance of $\nu_{yi}\left(t\right)$, the conservative and actual noise variances $R_{zyi}\left(t\right)$ and ${\bar{R}}_{zyi}\left(t\right)$ are given as follows
(34)$$\begin{array}{cc} R_{zyi}\left(t\right)= & \lambda_{\alpha i}\lambda_{\gamma 0i}\left(H_{i1}+H_{i2}J_{x_{2}}\right)X_{1}\left(t\right){\left(H_{i1}+H_{i2}J_{x_{2}}\right)}^{T}+\lambda_{\alpha i}\lambda_{\gamma i}\sum_{l=1}^{n_{a}}\sigma_{\alpha il}^{2}H_{il}X\left(t\right)H_{il}^{T}\\ \; & +\lambda_{\alpha i}\lambda_{\gamma i}H_{i2}U_{x_{2}}Q_{w}{\left(H_{i2}U_{x_{2}}\right)}^{T}+\lambda_{\alpha i}R_{i},\\ {\bar{R}}_{zyi}\left(t\right)= & \lambda_{\alpha i}\lambda_{\gamma 0i}\left(H_{i1}+H_{i2}J_{x_{2}}\right){\bar{X}}_{1}\left(t\right){\left(H_{i1}+H_{i2}J_{x_{2}}\right)}^{T}+\lambda_{\alpha i}\lambda_{\gamma i}\sum_{l=1}^{n_{a}}{\bar{\sigma }}_{\alpha il}^{2}H_{il}\bar{X}\left(t\right)H_{il}^{T}\\ \; & +\lambda_{\alpha i}\lambda_{\gamma i}H_{i2}U_{x_{2}}{\bar{Q}}_{w}{\left(H_{i2}U_{x_{2}}\right)}^{T}+\lambda_{\alpha i}{\bar{R}}_{i}. \end{array}$$

In (Equation 33) and (Equation 34), let
(35)$$\begin{array}{cc} \; & U_{1i}\left(t\right)=\sum_{l=1}^{n_{a}}\sigma_{\alpha il}^{2}H_{il}X\left(t\right)H_{il}^{T},\;U_{2i}\left(t\right)=\left(H_{i1}+H_{i2}J_{x_{2}}\right)X_{1}\left(t\right){\left(H_{i1}+H_{i2}J_{x_{2}}\right)}^{T},\\ \; & {\bar{U}}_{1i}\left(t\right)=\sum_{l=1}^{n_{a}}{\bar{\sigma }}_{\alpha il}^{2}H_{il}\bar{X}\left(t\right)H_{il}^{T},\;{\bar{U}}_{2i}\left(t\right)=\left(H_{i1}+H_{i2}J_{x_{2}}\right){\bar{X}}_{1}\left(t\right){\left(H_{i1}+H_{i2}J_{x_{2}}\right)}^{T},\\ \; & U_{3i}\left(t\right)=H_{i2}U_{x_{2}}Q_{w}{\left(H_{i2}U_{x_{2}}\right)}^{T},\;{\bar{U}}_{3i}\left(t\right)=H_{i2}U_{x_{2}}\left(t\right){\bar{Q}}_{w}{\left(H_{i2}U_{x_{2}}\right)}^{T}, \end{array}$$
then (Equation 33) and (Equation 34) can be simplified into the following equations
(36)$$\begin{array}{cc} R_{zi}\left(t\right) & =\lambda_{\gamma i}U_{1i}\left(t\right)+\lambda_{\gamma 0i}U_{2i}\left(t\right)+\lambda_{\gamma i}U_{3i}\left(t\right)+R_{i},\\ {\bar{R}}_{zi}\left(t\right) & =\lambda_{\gamma i}{\bar{U}}_{1i}\left(t\right)+\lambda_{\gamma 0i}{\bar{U}}_{2i}\left(t\right)+\lambda_{\gamma i}{\bar{U}}_{3i}\left(t\right)+{\bar{R}}_{i},\\ R_{zyi}\left(t\right) & =\lambda_{\alpha i}\lambda_{\gamma i}U_{1i}\left(t\right)+\lambda_{\alpha i}\lambda_{\gamma 0i}U_{2i}\left(t\right)+\lambda_{\alpha i}\lambda_{\gamma i}U_{3i}\left(t\right)+\lambda_{\alpha i}R_{i},\\ {\bar{R}}_{zyi}\left(t\right) & =\lambda_{\alpha i}\lambda_{\gamma i}{\bar{U}}_{1i}\left(t\right)+\lambda_{\alpha i}\lambda_{\gamma 0i}{\bar{U}}_{2i}\left(t\right)+\lambda_{\alpha i}\lambda_{\gamma i}{\bar{U}}_{3i}\left(t\right)+\lambda_{\alpha i}{\bar{R}}_{i}. \end{array}$$

Substituting (Equation 25) into $\omega_{ai}\left(t\right)$ in (Equation 27), we have
(37)$$\begin{array}{c} \omega_{ai}\left(t\right)=\Gamma_{ai}^{\left(1\right)}\sum_{u=1}^{4}C_{ui}\left(t\right)H_{cui}x_{ai}\left(t\right)+\omega_{ai}^{\left(1\right)}\left(t\right), \end{array}$$
where $\Gamma_{ai}^{\left(1\right)}={\left[0,\;\;0,\;\;I_{m1}\right]}^{T},$ $\omega_{ai}^{\left(1\right)}\left(t\right)={\left[\begin{array}{ccc} \omega_{t} & \nu_{zi}\left(t\right) & \nu_{zyi}^{\left(1\right)}\left(t\right) \end{array}\right]}^{T}$, $\nu_{zyi}^{\left(1\right)}\left(t\right)=\alpha_{i}\left(t\right)\gamma_{i}\left(t\right)\times \left(\sum_{l=1}^{n_{a}}\alpha_{il}\left(t\right)H_{il}\right)x\left(t\right)+\alpha_{i}\left(t\right)\gamma_{i}\left(t\right)H_{i2}U_{x_{2}}\omega \left(t\right)+\alpha_{i}\left(t\right)\nu_{i}\left(t\right)+\lambda_{\alpha i}\gamma_{0i}\left(t\right)\left(H_{i1}+H_{i2}J_{x_{2}}\right)x_{1}\left(t\right)$, we can obtain the conservative and actual variances $Q_{wai}^{\left(1\right)}\left(t\right)$ and ${\bar{Q}}_{wai}^{\left(1\right)}\left(t\right)$ as follows
(38)$$\begin{array}{c} Q_{wai}^{\left(1\right)}\left(t\right)=\left[\begin{array}{ccc} Q_{w} & \lambda_{\gamma i}Q_{w}{\left(H_{i2}U_{x2}\right)}^{T} & \lambda_{\alpha i}\lambda_{\gamma i}Q_{w}{\left(H_{i2}U_{x2}\right)}^{T}\\ \lambda_{\gamma i}\left(H_{i2}U_{x2}\right)Q_{w} & R_{zi}\left(t\right) & R_{zyi}\left(t\right)\\ \lambda_{\alpha i}\lambda_{\gamma i}\left(H_{i2}U_{x2}\right)Q_{w} & R_{zyi}{\left(t\right)}^{T} & \overset{˘}{R} \end{array}\right], \end{array}$$
(39)$$\begin{array}{c} {\bar{Q}}_{wai}^{\left(1\right)}\left(t\right)=\left[\begin{array}{ccc} {\bar{Q}}_{w} & \lambda_{\gamma i}{\bar{Q}}_{w}{\left(H_{i2}U_{x2}\right)}^{T} & \lambda_{\alpha i}\lambda_{\gamma i}{\bar{Q}}_{w}{\left(H_{i2}U_{x2}\right)}^{T}\\ \lambda_{\gamma i}\left(H_{i2}U_{x2}\right){\bar{Q}}_{w} & {\bar{R}}_{zi}\left(t\right) & {\bar{R}}_{zyi}\left(t\right)\\ \lambda_{\alpha i}\lambda_{\gamma i}\left(H_{i2}U_{x2}\right){\bar{Q}}_{w} & {\bar{R}}_{zyi}{\left(t\right)}^{T} & \overset{˘}{\bar{R}} \end{array}\right], \end{array}$$
where $\overset{˘}{R}=\lambda_{\alpha i}\lambda_{\gamma i}U_{1i}\left(t\right)+\lambda_{\alpha i}^{2}\lambda_{\gamma 0i}U_{2i}\left(t\right)+\lambda_{\alpha i}\lambda_{\gamma i}U_{3i}\left(t\right)+\lambda_{\alpha i}R_{i}$, $\overset{˘}{\bar{R}}=\lambda_{\alpha i}\lambda_{\gamma i}{\bar{U}}_{1i}\left(t\right)+\lambda_{\alpha i}^{2}\lambda_{\gamma 0i}{\bar{U}}_{2i}\left(t\right)+\lambda_{\alpha i}\lambda_{\gamma i}{\bar{U}}_{3i}\left(t\right)+\lambda_{\alpha i}{\bar{R}}_{i}$. Defining $U_{4i}\left(t\right)$ and ${\bar{U}}_{4i}\left(t\right)$, we have
(40)$$\begin{array}{c} Q_{wai}\left(t\right)=\Gamma_{ai}^{\left(1\right)}U_{4i}\left(t\right){\Gamma_{ai}^{\left(1\right)}}^{T}+Q_{wai}^{\left(1\right)}\left(t\right),\\ {\bar{Q}}_{wai}\left(t\right)=\Gamma_{ai}^{\left(1\right)}{\bar{U}}_{4i}\left(t\right){\Gamma_{ai}^{\left(1\right)}}^{T}+{\bar{Q}}_{wai}^{\left(1\right)}\left(t\right), \end{array}$$
where $U_{4i}\left(t\right)=\sum_{u=1}^{4}\sigma_{cui}^{2}H_{cui}X_{ai}\left(t\right)H_{cui}^{T},{\bar{U}}_{4i}\left(t\right)=\sum_{u=1}^{4}\sigma_{cui}^{2}H_{cui}{\bar{X}}_{ai}\left(t\right)H_{cui}^{T}$, then we have
(41)$$\begin{array}{c} R_{yi}\left(t\right)=\lambda_{\alpha i}\lambda_{\gamma i}U_{1i}\left(t\right)+\lambda_{\alpha i}^{2}\lambda_{\gamma 0i}U_{2i}\left(t\right)+\lambda_{\alpha i}R_{i}+\lambda_{\alpha i}\lambda_{\gamma i}U_{3i}\left(t\right)+U_{4i}\left(t\right),\\ {\bar{R}}_{yi}\left(t\right)=\lambda_{\alpha i}\lambda_{\gamma i}{\bar{U}}_{1i}\left(t\right)+\lambda_{\alpha i}^{2}\lambda_{\gamma 0i}{\bar{U}}_{2i}\left(t\right)+\lambda_{\alpha i}{\bar{R}}_{i}+\lambda_{\alpha i}\lambda_{\gamma i}{\bar{U}}_{3i}\left(t\right)+{\bar{U}}_{4i}\left(t\right), \end{array}$$
the conservative and actual cross-covariance $S_{ai}\left(t\right)$ and ${\bar{S}}_{ai}\left(t\right)$ are defined as follows
(42)$$\begin{array}{cc} S_{ai}\left(t\right) & =E\left[\omega_{ai}\left(t\right)\nu_{yj}^{T}\left(t\right)\right]=\left[\begin{array}{c} \lambda_{\alpha i}\lambda_{\gamma i}Q_{w}{\left(H_{i2}U_{x2}\right)}^{T}\\ R_{zyi}\left(t\right)\\ R_{yi}\left(t\right) \end{array}\right]\delta ij,\\ {\bar{S}}_{ai}\left(t\right) & =E\left[{\bar{\omega }}_{ai}\left(t\right){\bar{\nu }}_{yj}^{T}\left(t\right)\right]=\left[\begin{array}{c} \lambda_{\alpha i}\lambda_{\gamma i}{\bar{Q}}_{w}{\left(H_{i2}U_{x2}\right)}^{T}\\ {\bar{R}}_{zyi}\left(t\right)\\ {\bar{R}}_{yi}\left(t\right) \end{array}\right]\delta ij. \end{array}$$

**Lemma 1** ([28]). *(i) Let $A_{i}\geq 0,i=0,1,\cdots ,L$, then $\mathrm{diag}(A_{1},\cdots ,A_{L})\geq 0$. (ii) Let $A\geq 0,A\in R^{n\times n}$, and $A_{\delta }={\left(A_{ij}\right)}_{nL\times nL}$, $A_{ij}=A$, then $A_{\delta }\geq 0$. (iii) Let $A\geq 0,\;A\in R^{m\times m}$, then for arbitrary $C\in R^{p\times m}$, $CAC^{T}\geq 0$.*

Parameters $\Delta X_{1}\left(t\right)$, ΔX(t), $\Delta X_{ai}\left(t\right)$, $\Delta R_{yi}\left(t\right)$ and $\Delta Q_{wai}\left(t\right)$ are defined as $\Delta X_{1}\left(t\right)=X_{1}\left(t\right)-{\bar{X}}_{1}\left(t\right)$, $\Delta X\left(t\right)=X\left(t\right)-\bar{X}\left(t\right)$, $\Delta X_{ai}\left(t\right)=X_{1}\left(t\right)-{\bar{X}}_{ai}\left(t\right)$, $\Delta R_{yi}\left(t\right)=R_{yi}\left(t\right)-{\bar{R}}_{yi}\left(t\right)$, $\Delta Q_{wai}\left(t\right)=Q_{wai}\left(t\right)-{\bar{Q}}_{wai}\left(t\right)$.

**Theorem 1.** 
*For all admissible uncertain variance ${\bar{Q}}_{\omega }$, ${\bar{R}}_{i}$, ${\bar{\sigma }}_{il}$ in (Equation 13), all of the following inequalities are true, that is,*

(43)
$$\Delta X_{1}\left(t\right)\geq 0,\;\;\Delta X\left(t\right)\geq 0,\;\;\Delta X_{ai}\left(t\right)\geq 0,\;\;\Delta R_{yi}\left(t\right)\geq 0,\;\;\Delta Q_{wai}\left(t\right)\geq 0.$$



**Proof of Theorem 1.** From (Equation 31) and (Equation 32), it is easy to obtain
(44)$$\Delta X_{1}\left(t+1\right)=J_{x1}\Delta X_{1}\left(t\right)J_{x1}^{T}+U_{x1}\Delta Q_{w}U_{x1}^{T},$$
with the initial condition $\Delta X_{1}\left(0\right)=X_{1}\left(0\right)-X_{1}\left(0\right)\geq 0$, $\Delta Q_{w}\geq 0$, applying Lemma 1, iterating (Equation 44) yield $\Delta X_{1}\left(t\right)\geq 0$.Let $Q=[Q_{1},\;\;Q_{2}]$, from (Equation 17), (Equation 19) and (20), it is easy to obtain
(45)$$x\left(t\right)=\left(Q_{1}+Q_{2}J_{x2}\right)x_{1}\left(t\right)+Q_{2}U_{x2}\omega \left(t\right)+Q_{2}G_{x2}u\left(t\right),$$
(46)$$\Delta X\left(t\right)=\left(Q_{1}+Q_{2}J_{x2}\right)\Delta X_{1}\left(t\right){\left(Q_{1}+Q_{2}J_{X2}\right)}^{T}+Q_{2}U_{x2}\Delta Q_{w}{\left(Q_{2}U_{x2}\right)}^{T},$$
because of $\Delta X_{1}\left(t\right)\geq 0$ and $\Delta Q_{w}\geq 0$, based on Lemma 1, we have ΔX(t)≥0.Rewriting $Q_{wai}^{\left(1\right)}\left(t\right)$ as follows
(47)$$\begin{array}{cc} \Delta Q_{wai}^{\left(1\right)}\left(t\right)= & D^{\left(0\right)}\left[\begin{array}{ccc} \Delta Q_{w} & \Delta Q_{w} & \Delta Q_{w}\\ \Delta Q_{w} & 0 & 0\\ \Delta Q_{w} & 0 & 0 \end{array}\right]{D^{\left(0\right)}}^{T}+\lambda_{\alpha i}\lambda_{\gamma i}D^{\left(1\right)}\Delta U_{1i}{D^{\left(1\right)}}^{T}\\ \; & +\lambda_{\gamma 0i}D^{\left(2\right)}\Delta U_{2i}{D^{\left(2\right)}}^{T}+\lambda_{\alpha i}\lambda_{\gamma i}D^{\left(3\right)}\Delta U_{3i}{D^{\left(3\right)}}^{T}+\lambda_{\alpha i}D^{\left(4\right)}\Delta R_{i}{D^{\left(4\right)}}^{T}\\ \; & +\left[\begin{array}{ccc} 0 & 0 & 0\\ 0 & \lambda_{\gamma i}\left(1-\lambda_{\alpha i}\right)\Delta U_{1i}+\lambda_{\gamma i}\left(1-\lambda_{\alpha i}\right)\Delta U_{3i}+\left(1-\lambda_{\alpha i}\right)\Delta R_{i} & 0\\ \Delta 0 & 0 & 0 \end{array}\right] \end{array}$$
where $D^{\left(0\right)}=\mathrm{diag}\left\{I_{n1},\;\lambda_{\gamma i}H_{i1}U_{x2},\;\lambda_{\alpha i}\lambda_{\gamma i}H_{i1}U_{x2}\right\},\;D^{\left(1\right)}=D^{\left(3\right)}=D^{\left(4\right)}=\left[\begin{array}{c} 0\\ I_{m1}\\ I_{m1} \end{array}\right],D^{\left(2\right)}=\mathrm{diag}\left\{I_{n1},\;I_{m1},\;\lambda_{\alpha i}I_{m1}\right\}$.Let $\Delta U_{1i}=U_{1i}-{\bar{U}}_{1i}$, $\Delta U_{2i}=U_{2i}-{\bar{U}}_{2i}$, $\Delta U_{3i}=U_{3i}-{\bar{U}}_{3i}$, from (Equation 35), we have
(48)$$\begin{array}{cc} \Delta U_{1i} & =\sum_{l=1}^{n_{a}}\left(\Delta \sigma_{\alpha il}^{2}H_{il}X\left(t\right)H_{il}^{T}+\sigma_{\alpha il}^{2}H_{il}\Delta X\left(t\right)H_{il}^{T}\right),\\ \Delta U_{2i} & =\left(H_{i1}+H_{i2}J_{x_{2}}\right)\Delta X_{1}\left(t\right){\left(H_{i1}+H_{i2}J_{x_{2}}\right)}^{T},\\ \Delta U_{3i} & =H_{i2}U_{x_{2}}\Delta Q_{w}{\left(H_{i2}U_{x_{2}}\right)}^{T}, \end{array}$$
since $\Delta \sigma_{\alpha il}^{2}\geq 0$, ΔX(t)≥0, $\Delta X_{1}\left(t\right)\geq 0$ and $\Delta Q_{w}\geq 0$, based on Lemma 1, it is easy to obtain
(49)$$\begin{array}{c} \Delta U_{1i}\geq 0,\;\;\Delta U_{2i}\geq 0,\;\;\Delta U_{3i}\geq 0, \end{array}$$
applying Lemma 1, we can easily obtain $\Delta Q_{wai}^{\left(1\right)}\left(t\right)\geq 0$, from (Equation 31), (Equation 32) and (Equation 40), it is easy to obtain $\Delta X_{ai}\left(t+1\right)$ as follows
(50)$$\Delta X_{ai}\left(t+1\right)=\Phi_{ai}\Delta X_{ai}\left(t\right)\Phi_{ai}^{T}+\Gamma_{ai}\Delta Q_{wai}\left(t\right)\Gamma_{ai}^{T}$$
(51)$$\Delta X_{ai}\left(t+1\right)=\Phi_{ai}\Delta X_{ai}\left(t\right)\Phi_{ai}^{T}+\Gamma_{ai}\left(\Gamma_{ai}^{\left(1\right)}\Delta U_{4i}\left(t\right){\Gamma_{ai}^{\left(1\right)}}^{T}+\Delta Q_{wai}^{\left(1\right)}\left(t\right)\right)\Gamma_{ai}^{T},$$
then we have
(52)$$\begin{array}{c} \Delta U_{4i}\left(t\right)=\sum_{u=1}^{4}\sigma_{cui}^{2}H_{cui}\Delta X_{ai}\left(t\right)H_{cui}^{T}, \end{array}$$
we can easily obtain $\Delta U_{4i}\left(t\right)\geq 0$, with the initial condition $\Delta X_{ai}\left(0\right)\geq 0$. According to (Equation 50) and applying mathematical induction, yield $\Delta X_{ai}\left(t\right)\geq 0$, since $\Delta Q_{wai}^{\left(1\right)}\left(t\right)\geq 0$, $\Delta U_{4i}\left(t\right)\geq 0$, from (Equation 40), yield $\Delta Q_{wai}\left(t\right)\geq 0$.From (Equation 41), it is easy to obtain
(53)$$\begin{array}{c} \Delta R_{yi}\left(t\right)=\lambda_{\alpha i}\lambda_{\gamma i}\Delta U_{1i}+\lambda_{\alpha i}\Delta R_{i}+\lambda_{\alpha i}^{2}\lambda_{\gamma 0i}\Delta U_{2i}+\lambda_{\alpha i}\lambda_{\gamma i}\Delta U_{3i}, \end{array}$$
from (Equation 49), yield $\Delta R_{yi}\left(t\right)\geq 0$. The Proof of Theorem 1 is completed. □

## 4. Robust Kalman Estimator of Descriptor System

### 4.1. Conservative Kalman Estimator of New State Space Model

For the new standard system (Equation 28) and (29), applying the Kalman filtering algorithm [29] yields the optimal Kalman estimator ${\hat{x}}_{ai}\left(t|t+N\right)$ (include filter (N=0), predictor (N=−1), smoother (N≥1))
(54)$$\begin{array}{cc} \; & {\hat{x}}_{ai}\left(t+1|t\right)=\Psi_{pi}\left(t\right){\hat{x}}_{ai}\left(t|t-1\right)+K_{pi}\left(t\right)y_{ai}\left(t\right)+G_{ai}\left(t\right), \end{array}$$
(55)$$\begin{array}{cc} \; & {\hat{x}}_{ai}\left(t|t+N\right)=hatx_{ai}\left(t|t-1\right)+\sum_{r=0}^{N}K_{i}\left(t|t+r\right)\varepsilon_{i}\left(t+r\right),\;N\geq 0, \end{array}$$
(56)$$\begin{array}{cc} \; & \varepsilon_{i}\left(t\right)=y_{ai}\left(t\right)-H_{ai}{\hat{x}}_{ai}\left(t|t-1\right), \end{array}$$
where $\Psi_{pi}\left(t\right)=\Phi_{ai}-K_{pi}\left(t\right)H_{ai},\;K_{pi}\left(t\right)=\left(\Phi_{ai}P_{ai}\left(t|t-1\right)H_{ai}^{T}+\Gamma_{ai}S_{ai}\left(t\right)\right)Q_{\varepsilon i}^{-1}\left(t\right),\;Q_{\varepsilon i}\left(t\right)=H_{ai}P_{ai}\left(t|t-1\right)H_{ai}^{T}+R_{yi}\left(t\right),\;K_{i}\left(t|t\right)=P_{ai}\left(t|t-1\right)H_{ai}^{T}Q_{\varepsilon i}^{-1}\left(t\right),K_{i}\left(t|t+r\right)=P_{ai}\left(t|t-1\right)\times \left\{\prod_{j=0}^{r-1}\Psi_{pi}\left(t+j\right)\right\}H_{ai}^{T}\varepsilon_{i}\left(t+r\right),\;r\geq 1$, and the conservative prediction error variance satisfies the Riccati equation
(57)$$\begin{array}{cc} P_{ai}\left(t+1|t\right)= & \Phi_{ai}P_{ai}\left(t|t-1\right)\Phi_{ai}^{T}-\left(\Phi_{ai}P_{ai}\left(t|t-1\right)H_{ai}^{T}+\Gamma_{ai}S_{ai}\left(t\right)\right)(H_{ai}P_{ai}\left(t|t-1\right)H_{ai}^{T}\\ \; & +R_{yi}{\left(t\right))}^{-1}\times {\left(\Phi_{ai}P_{ai}\left(t|t-1\right)H_{ai}^{T}+\Gamma_{ai}S_{ai}\left(t\right)\right)}^{T}+\Gamma_{ai}Q_{wai}\Gamma_{ai}^{T}. \end{array}$$

The one-step predicting error is defined as ${\tilde{x}}_{ai}\left(t+1|t\right)=x_{ai}\left(t\right)-{\hat{x}}_{ai}\left(t+1|t\right)$
(58)$$\begin{array}{c} {\tilde{x}}_{ai}\left(t+1|t\right)=\Psi_{pi}\left(t\right){\tilde{x}}_{ai}\left(t|t-1\right)+\left[\Gamma_{ai},\;-K_{pi}\left(t\right)\right]\xi_{wv}\left(t\right), \end{array}$$
where $\xi_{wv}\left(t\right)=\left[\begin{array}{c} \omega_{ai}\left(t\right)\\ \nu_{yi}\left(t\right) \end{array}\right]$.

Furthermore, the conservative and the actual variance $\Lambda_{i}\left(t\right)$ and ${\bar{\Lambda }}_{i}\left(t\right)$ are defined as follows
(59)$$\begin{array}{c} \Lambda_{i}\left(t\right)=\left[\begin{array}{cc} Q_{wai}\left(t\right) & S_{ai}\left(t\right)\\ S_{ai}^{T}\left(t\right) & R_{yi}\left(t\right) \end{array}\right],\;\;{\bar{\Lambda }}_{i}\left(t\right)=\left[\begin{array}{cc} {\bar{Q}}_{wai}\left(t\right) & {\bar{S}}_{ai}\left(t\right)\\ {\bar{S}}_{ai}^{T}\left(t\right) & {\bar{R}}_{yi}\left(t\right) \end{array}\right]. \end{array}$$

The conservative and actual one-step prediction error variance $P_{ai}\left(t+1|t\right)$ and ${\bar{P}}_{ai}\left(t+1|t\right)$ can be rewritten as the following Lyapunov function
(60)$$\begin{array}{c} P_{ai}\left(t+1|t\right)=\Psi_{pi}\left(t\right)P_{ai}\left(t|t-1\right)\Psi_{pi}^{T}\left(t\right)+\left[\Gamma_{ai},\;-K_{pi}\left(t\right)\right]\Lambda_{i}\left(t\right){\left[\Gamma_{ai},\;-K_{pi}\left(t\right)\right]}^{T} \end{array}$$
(61)$$\begin{array}{c} {\bar{P}}_{ai}\left(t+1|t\right)=\Psi_{pi}\left(t\right){\bar{P}}_{ai}\left(t|t-1\right)\Psi_{pi}^{T}\left(t\right)+\left[\Gamma_{ai},\;-K_{pi}\left(t\right)\right]{\bar{\Lambda }}_{i}\left(t\right){\left[\Gamma_{ai},\;-K_{pi}\left(t\right)\right]}^{T}, \end{array}$$
with the initial values $P_{ai}\left(1|0\right)=\mathrm{diag}\left\{P_{01},\;\;0,\;\;0\right\},{\bar{P}}_{ai}\left(1|0\right)=\mathrm{diag}\left\{{\bar{P}}_{01},\;\;0,\;\;0\right\}$.

From (29), (56), we have $\varepsilon_{i}\left(t+r\right)=H_{ai}{\tilde{x}}_{ai}\left(t+r|t+r-1\right)+\nu_{yi}\left(t+r\right)$, iterating (Equation 58), we can obtain
(62)$$\begin{array}{c} {\tilde{x}}_{ai}\left(t|t+N\right)=\Psi_{N}\left(t\right){\tilde{x}}_{ai}\left(t|t-1\right)+\sum_{r=0}^{N}\left[K_{r}^{Nw}\left(t\right),\;\;K_{r}^{Nv}\left(t\right)\right]\xi_{wv}\left(t+r\right), \end{array}$$
where $\Psi_{pi}\left(t+r|t\right)=\Psi_{pi}\left(t+r-1\right)\cdots \Psi_{pi}\left(t\right)$ and $\Psi_{pi}\left(t|t\right)=I_{n1+2mi}$,
$$\begin{array}{c} \Psi_{N}\left(t\right)=I_{n1+2mi}-\sum_{r=0}^{N}K_{i}\left(t|t+r\right)H_{ai}\Psi_{pi}\left(t+r|t\right), \end{array}$$
$$\begin{array}{c} \left\{\begin{array}{cc} K_{r}^{Nw}\left(t\right)=-\sum_{j=r+1}^{N}K_{i}\left(t+r\right)H_{ai}\Psi_{pi}\left(t+j|t+r+1\right)\Gamma_{ai}, & \\ K_{r}^{Nw}\left(t\right)=0,\;N\geq 0, \end{array}\right. \end{array}$$
$$\begin{array}{c} \left\{\begin{array}{cc} K_{r}^{Nv}\left(t\right)=-\sum_{j=r+1}^{N}K_{i}\left(t+r\right)H_{ai}\Psi_{pi}\left(t+j|t+r+1\right)K_{pi}\left(t+r\right)-K_{i}\left(t|t+r\right), & \\ K_{r}^{Nv}\left(t\right)=-K_{i}\left(t|t+r\right),\;N\geq 0. \end{array}\right. \end{array}$$

Furthermore, the optimal conservative white noise deconvolution estimator ${\hat{\omega }}_{ai}\left(t|t+N\right)$ of fictitious noise $\omega_{ai}\left(t\right)$ is
(63)$$\begin{array}{cc} \; & {\hat{\omega }}_{ai}\left(t|t-1\right)=0, \end{array}$$
(64)$$\begin{array}{cc} \; & {\hat{\omega }}_{ai}\left(t|t+N\right)=\sum_{r=0}^{N}M_{wi}\left(t|t+r\right)\varepsilon_{i}\left(t+r\right),\;N\geq 0, \end{array}$$
where $M_{wi}\left(t|t\right)=S_{ai}\left(t\right)Q_{\varepsilon i}^{-1}\left(t\right),M_{wi}\left(t|t\right)=\left(Q_{wai}\left(t\right)\Gamma_{ai}^{T}-S_{ai}\left(t\right)K_{pi}^{T}\left(t\right)\right)\left\{\prod_{j=0}^{r-1}\Psi_{pi}^{T}\left(t+j\right)\right\}\times H_{ai}^{T}Q_{\varepsilon i}^{-1}\left(t+r\right)$, noise estimation error is defined as ${\tilde{\omega }}_{ai}\left(t|t+N\right)=\omega_{ai}\left(t\right)-{\hat{\omega }}_{ai}\left(t|t+N\right)$, then it is easy to obtain
(65)$$\begin{array}{c} {\tilde{\omega }}_{ai}\left(t|t+N\right)=\Psi_{N}^{w}\left(t\right){\tilde{x}}_{ai}\left(t|t-1\right)+\sum_{r=0}^{N}\left[M_{r}^{Nw}\left(t\right),\;\;M_{r}^{Nv}\left(t\right)\right]\xi_{wv}\left(t+r\right), \end{array}$$
where $\Psi_{N}^{w}\left(t\right)=-\sum_{r=0}^{N}M_{wi}\left(t|t+r\right)H_{ai}\Psi_{pi}\left(t+r|t\right)$,
$$\begin{array}{c} \left\{\begin{array}{cc} M_{0}^{Nw}\left(t\right)=I_{r}-\sum_{k=1}^{N}M_{\omega i}\left(t|t+k\right)H_{ai}\Psi_{pi}\left(t+k|t+1\right)\Gamma_{ai}, & \\ M_{r}^{Nw}\left(t\right)=-\sum_{j=r+1}^{N}M_{\omega i}\left(t|t+j\right)H_{ai}\Psi_{pi}\left(t+j|t+r+1\right)\Gamma_{ai},\;\;r=0,\cdots ,\;N-1 & \\ M_{N}^{Nw}\left(t\right)=0, \end{array}\right. \end{array}$$
$$\begin{array}{c} \left\{\begin{array}{c} M_{r}^{Nv}\left(t\right)=\sum_{j=r+1}^{N}M_{\omega i}\left(t|t+j\right)H_{ai}\Psi_{pi}\left(t+j|t+r+1\right)K_{pi}\left(t+r\right)-M_{\omega i}\left(t|t+r\right),\;\;\\ M_{r}^{Nv}\left(t\right)=-M_{\omega i}\left(t|t+N\right). \end{array}\right. \end{array}$$

The conservative and actual estimation error variances $P_{ai}\left(t|t+N\right)$ and ${\bar{P}}_{ai}\left(t|t+N\right)$ are defined as follows
(66)$$\begin{array}{cc} P_{ai}\left(t|t+N\right)= & \Psi_{N}\left(t\right)P_{ai}\left(t|t-1\right)\Psi_{N}^{T}\left(t\right)\\ \; & +\sum_{r=0}^{N}\left[K_{r}^{Nw}\left(t\right),\;\;K_{r}^{Nv}\left(t\right)\right]\Lambda_{i}\left(t+r\right){\left[K_{r}^{Nw}\left(t\right),\;\;K_{r}^{Nv}\left(t\right)\right]}^{T},\\ {\bar{P}}_{ai}\left(t|t+N\right)= & \Psi_{N}\left(t\right){\bar{P}}_{ai}\left(t|t-1\right)\Psi_{N}^{T}\left(t\right)\\ \; & +\sum_{r=0}^{N}\left[K_{r}^{Nw}\left(t\right),\;\;K_{r}^{Nv}\left(t\right)\right]{\bar{\Lambda }}_{i}\left(t+r\right){\left[K_{r}^{Nw}\left(t\right),\;\;K_{r}^{Nv}\left(t\right)\right]}^{T}. \end{array}$$

The conservative and actual estimation error variances $P_{wai}\left(t|t+N\right)$ and ${\bar{P}}_{wai}\left(t|t+N\right)$ of ${\tilde{\omega }}_{ai}\left(t|t+N\right)$ are defined as follows
(67)$$\begin{array}{cc} P_{wai}\left(t|t+N\right)= & \Psi_{N}^{w}\left(t\right)P_{ai}\left(t|t-1\right){\Psi_{N}^{w}}^{T}\left(t\right)\\ \; & +\sum_{r=0}^{N}\left[M_{r}^{Nw}\left(t\right),\;\;M_{r}^{Nv}\left(t\right)\right]\Lambda_{i}\left(t+r\right){\left[M_{r}^{Nw}\left(t\right),\;\;M_{r}^{Nv}\left(t\right)\right]}^{T},\\ {\bar{P}}_{wai}\left(t|t+N\right)= & \Psi_{N}^{w}\left(t\right){\bar{P}}_{ai}\left(t|t-1\right){\Psi_{N}^{w}}^{T}\left(t\right)\\ \; & +\sum_{r=0}^{N}\left[M_{r}^{Nw}\left(t\right),\;\;M_{r}^{Nv}\left(t\right)\right]{\bar{\Lambda }}_{i}\left(t+r\right){\left[M_{r}^{Nw}\left(t\right),\;\;M_{r}^{Nv}\left(t\right)\right]}^{T}. \end{array}$$

### 4.2. Conservative Kalman Estimator of Original Descriptor System

**Theorem 2.** 
*For the uncertain MSDS (1)–(4) with Assumptions 1–3, the robust Kalman estimator $\hat{x}\left(t|t+N\right)$ is obtained as follows*

(68)
$$\begin{array}{c} \hat{x}\left(t|t+N\right)=Q_{0}\left[\begin{array}{c} {\hat{x}}_{ai}\left(t|t+N\right)\\ {\hat{\omega }}_{ai}\left(t|t+N\right) \end{array}\right]+Q\left[\begin{array}{c} 0\\ G_{x2} \end{array}\right]u\left(t\right), \end{array}$$

*where*

$$\begin{array}{c} Q_{0}=Q\left[\begin{array}{cc} I_{n1} & 0\\ J_{x1} & U_{x2} \end{array}\right]\left[\begin{array}{cccccc} I_{n_{1}} & 0_{n_{1}\times m_{i}} & 0_{n_{1}\times m_{i}} & 0_{n_{1}\times n_{w}} & 0_{n_{1}\times n_{w}} & 0_{n_{1}\times n_{w}}\\ 0_{n_{w}\times n_{1}} & 0_{n_{w}\times m_{i}} & 0_{n_{w}\times m_{i}} & I_{n_{w}} & 0_{n_{w}\times m_{i}} & 0_{n_{w}\times m_{i}}, \end{array}\right], \end{array}$$


$$\begin{array}{cc} \; & P\left(t|t+N\right)=Q_{0}\left[\begin{array}{cc} P_{ai}\left(t|t+N\right) & P_{x\omega }\left(t|t+N\right)\\ P_{x\omega }^{T}\left(t|t+N\right) & P_{wai}\left(t|t+N\right) \end{array}\right],\\ \; & \bar{P}\left(t|t+N\right)=Q_{0}\left[\begin{array}{cc} {\bar{P}}_{ai}\left(t|t+N\right) & {\bar{P}}_{x\omega }\left(t|t+N\right)\\ {\bar{P}}_{x\omega }^{T}\left(t|t+N\right) & {\bar{P}}_{wai}\left(t|t+N\right) \end{array}\right], \end{array}$$

*where*

$$\begin{array}{cc} P_{x\omega }\left(t|t+N\right)= & \Psi_{N}\left(t\right)P_{ai}\left(t|t-1\right){\Psi_{N}^{w}}^{T}\left(t\right)\\ \; & +\sum_{r=0}^{N}\left[K_{r}^{Nw}\left(t\right),\;\;K_{r}^{Nv}\left(t\right)\right]\Lambda_{i}\left(t+r\right){\left[M_{r}^{Nw}\left(t\right),\;\;M_{r}^{Nv}\left(t\right)\right]}^{T},\\ {\bar{P}}_{x\omega }\left(t|t+N\right)= & \Psi_{N}\left(t\right){\bar{P}}_{ai}\left(t|t-1\right){\Psi_{N}^{w}}^{T}\left(t\right)\\ \; & +\sum_{r=0}^{N}\left[K_{r}^{Nw}\left(t\right),\;\;K_{r}^{Nv}\left(t\right)\right]{\bar{\Lambda }}_{i}\left(t+r\right){\left[M_{r}^{Nw}\left(t\right),\;\;M_{r}^{Nv}\left(t\right)\right]}^{T}. \end{array}$$



**Proof of Theorem 2.** From (Equation 27), we can obtain
(69)$$\begin{array}{cc} \; & x_{1}\left(t\right)=\left[I_{n_{1}},\;\;0_{n_{1}\times m_{i}},\;\;0_{n_{1}\times m_{i}}\right]x_{ai}\left(t\right), \end{array}$$
(70)$$\begin{array}{cc} \; & \omega \left(t\right)=\left[I_{n_{w}},\;\;0_{n_{w}\times m_{i}},\;\;0_{n_{w}\times m_{i}}\right]\omega_{ai}\left(t\right). \end{array}$$Substituting (Equation 69) and (70) into (Equation 17) yields
(71)$$\begin{array}{cc} x(t)= & Q\left[\begin{array}{c} x_{1}\left(t\right)\\ J_{x1}x_{1}\left(t\right)+U_{x2}\omega \left(t\right)+G_{x2}u\left(t\right) \end{array}\right]\\ = & Q\left[\begin{array}{cc} I_{n1} & 0\\ J_{x1} & U_{x2} \end{array}\right]\left[\begin{array}{cccccc} I_{n_{1}} & 0_{n_{1}\times m_{i}} & 0_{n_{1}\times m_{i}} & 0_{n_{1}\times n_{w}} & 0_{n_{1}\times n_{w}} & 0_{n_{1}\times n_{w}}\\ 0_{n_{w}\times n_{1}} & 0_{n_{w}\times m_{i}} & 0_{n_{w}\times m_{i}} & I_{n_{w}} & 0_{n_{w}\times m_{i}} & 0_{n_{w}\times m_{i}} \end{array}\right]\left[\begin{array}{c} {\hat{x}}_{ai}\left(t|t+N\right)\\ {\hat{\omega }}_{ai}\left(t|t+N\right) \end{array}\right]\\ \; & +\left[\begin{array}{c} 0\\ G_{x2} \end{array}\right]u\left(t\right). \end{array}$$Taking the projection of (Equation 71), we can obtain (Equation 68). Subtracting (Equation 71) from (Equation 68), yields
(72)$$\begin{array}{c} \tilde{x}\left(t|t+N\right)=Q_{0}\left[\begin{array}{c} {\tilde{x}}_{ai}\left(t|t+N\right)\\ {\tilde{\omega }}_{ai}\left(t|t+N\right) \end{array}\right], \end{array}$$
then we can obtain the conservative and actual state estimation error variance $E[\tilde{x}\left(t|t+N\right){\tilde{x}}^{T}\left(t|t+N\right)]$. The Proof of Theorem 2 is completed. □

## 5. Robust Analysis

**Theorem 3.** 
*Consider the uncertain MSDS (Equation 1)–(4), for all admissible uncertain variance ${\bar{Q}}_{\omega }$, ${\bar{R}}_{i}$, ${\bar{\sigma }}_{il}$ in (Equation 13), $\Delta \Lambda_{i}\left(t\right)\geq 0$, and the actual estimation error variance $\bar{P}\left(t|t+N\right)$ has upper bound P(t|t+N), and the trace of error variance $\mathrm{tr}(\bar{P}\left(t|t+N\right))$ has upper bound tr(P(t|t+N)), that is*

(73)
$$\begin{array}{c} \Delta P(t|t+N)\geq 0,\;\;\Delta \mathrm{tr}(P(t|t+N))\geq 0, \end{array}$$

*where $\Delta \Lambda_{i}\left(t\right)=\Lambda_{i}\left(t\right)-{\bar{\Lambda }}_{i}\left(t\right)$, $\Delta P\left(t|t+N\right)=P\left(t|t+N\right)-\bar{P}\left(t|t+N\right)$, $\Delta \mathrm{tr}\left(P\left(t|t+N\right)\right)=\mathrm{tr}\left(P\left(t|t+N\right)\right)-\mathrm{tr}\left(\bar{P}\left(t|t+N\right)\right)$.*


**Proof of Theorem 3.** According to (Equation 59), it is easy to obtain
(74)$$\begin{array}{c} \Delta \Lambda_{i}\left(t\right)=\left[\begin{array}{cc} \Delta Q_{wai}\left(t\right) & \Delta S_{ai}\left(t\right)\\ \Delta S_{ai}^{T}\left(t\right) & \Delta R_{yi}\left(t\right) \end{array}\right], \end{array}$$
rewriting $\Delta \Lambda_{i}\left(t\right)$ as follows
(75)$$\begin{array}{c} \Delta \Lambda_{i}\left(t\right)=\Delta \Lambda_{i}^{\left(1\right)}\left(t\right)+\Delta \Lambda_{i}^{\left(2\right)}\left(t\right), \end{array}$$
where
$$\begin{array}{c} \Delta \Lambda_{i}^{\left(1\right)}\left(t\right)=\left[\begin{array}{cc} \Delta Q_{wai}\left(t\right) & 0\\ 0 & \Delta R_{yi}\left(t\right) \end{array}\right],\;\;\Delta \Lambda_{i}^{\left(2\right)}\left(t\right)=\left[\begin{array}{cc} 0 & \Delta S_{ai}\left(t\right)\\ \Delta S_{ai}^{T}\left(t\right) & 0 \end{array}\right] \end{array}$$
$$\begin{array}{cc} \Delta \Lambda_{i}^{\left(2\right)}\left(t\right)= & \left[\begin{array}{cccc} 0 & 0 & 0 & \lambda_{\alpha i}\Delta Q_{w}{\left(H_{i2}U_{x2}\right)}^{T}\\ 0 & 0 & 0 & \Delta R_{zyi}\left(t\right)\\ 0 & 0 & 0 & \Delta R_{yi}\left(t\right)\\ \lambda_{\alpha i}H_{i2}U_{x2}\Delta Q_{w} & \Delta R_{zyi}\left(t\right) & \Delta R_{yi}\left(t\right) & 0 \end{array}\right]\\ = & \lambda_{\alpha i}\mathrm{diag}\left\{I_{n_{i}\times m_{1}},I_{m_{i}},I_{m_{i}},H_{i2}U_{x2}\right\}\left[\begin{array}{cccc} 0 & 0 & 0 & \Delta Q_{w}\\ 0 & 0 & 0 & 0\\ 0 & 0 & 0 & 0\\ \Delta Q_{w} & 0 & 0 & 0 \end{array}\right]\\ \; & \times \mathrm{diag}{\left\{I_{n_{i}\times m_{1}},I_{m_{i}},I_{m_{i}},H_{i2}U_{x2}\right\}}^{T}+\left[\begin{array}{cccc} 0 & 0 & 0 & 0\\ 0 & 0 & 0 & \Delta R_{zyi}\left(t\right)\\ 0 & 0 & 0 & 0\\ 0 & \Delta R_{zyi}\left(t\right) & 0 & 0 \end{array}\right]\\ \; & +\left[\begin{array}{cccc} 0 & 0 & 0 & 0\\ 0 & 0 & 0 & 0\\ 0 & 0 & 0 & \Delta R_{yi}\left(t\right)\\ 0 & 0 & \Delta R_{yi}\left(t\right) & 0 \end{array}\right], \end{array}$$
since $\Delta Q_{wai}\left(t\right)\geq 0$, $\Delta R_{yi}\left(t\right)\geq 0$, we have $\Delta \Lambda_{i}^{\left(1\right)}\left(t\right)\geq 0$, applying Lemma 1, we have $\Delta \Lambda_{i}^{\left(2\right)}\left(t\right)\geq 0$, then $\Delta \Lambda_{i}\left(t\right)\geq 0$.Parameters $\Delta P_{ai}\left(t|t-1\right)$, $\Delta P_{ai}\left(t|t+N\right)$, $\Delta P_{wai}\left(t|t+N\right)$ and $\Delta P_{xw}\left(t|t+N\right)$ are defined as follows:
(76)$$\begin{array}{cc} \; & \Delta P_{ai}\left(t|t-1\right)=P_{ai}\left(t|t-1\right)-{\bar{P}}_{ai}\left(t|t-1\right),\\ \; & \Delta P_{ai}\left(t|t+N\right)=P_{ai}\left(t|t+N\right)-{\bar{P}}_{ai}\left(t|t+N\right),\\ \; & \Delta P_{wai}\left(t|t+N\right)=P_{wai}\left(t|t+N\right)-{\bar{P}}_{wai}\left(t|t+N\right),\\ \; & \Delta P_{xw}\left(t|t+N\right)=P_{ai}\left(t|t+N\right)-{\bar{P}}_{xw}\left(t|t+N\right), \end{array}$$
from (Equation 60) and (61) and (Equation 67), it is easy to obtain
(77)$$\begin{array}{cc} \Delta P_{ai}\left(t|t-1\right)= & \Psi_{pi}\left(t\right)\Delta P_{ai}\left(t|t-1\right)\Psi_{pi}^{T}\left(t\right)+\left[\Gamma_{ai},\;-K_{pi}\left(t\right)\right]\Delta \Lambda_{i}\left(t\right){\left[\Gamma_{ai},\;-K_{pi}\left(t\right)\right]}^{T},\\ \Delta P_{ai}\left(t|t+N\right)= & \Psi_{N}\left(t\right)\Delta P_{ai}\left(t|t-1\right)\Psi_{N}^{T}\left(t\right)\\ \; & +\sum_{r=0}^{N}\left[K_{r}^{Nw}\left(t\right),\;\;K_{r}^{Nv}\left(t\right)\right]\Delta \Lambda_{i}\left(t+r\right){\left[K_{r}^{Nw}\left(t\right),\;\;K_{r}^{Nv}\left(t\right)\right]}^{T},\\ \Delta P_{wai}\left(t|t+N\right)= & \Psi_{N}^{w}\left(t\right)\Delta P_{ai}\left(t|t-1\right){\Psi_{N}^{w}}^{T}\left(t\right)\\ \; & +\sum_{r=0}^{N}\left[M_{r}^{Nw}\left(t\right),\;\;M_{r}^{Nv}\left(t\right)\right]\Delta \Lambda_{i}\left(t+r\right){\left[M_{r}^{Nw}\left(t\right),\;\;M_{r}^{Nv}\left(t\right)\right]}^{T},\\ \Delta P_{x\omega }\left(t|t+N\right)= & \Psi_{N}\left(t\right)\Delta P_{ai}\left(t|t-1\right){\Psi_{N}^{w}}^{T}\left(t\right) \end{array}$$
(78)$$\begin{array}{cc} \; & +\sum_{r=0}^{N}\left[K_{r}^{Nw}\left(t\right),\;\;K_{r}^{Nv}\left(t\right)\right]\Delta \Lambda_{i}\left(t+r\right){\left[M_{r}^{Nw}\left(t\right),\;\;M_{r}^{Nv}\left(t\right)\right]}^{T}, \end{array}$$
with the initial condition $\Delta P_{ai}\left(1|0\right)\geq 0$, applying mathematical induction method, yield
(79)$$\begin{array}{c} \Delta P_{ai}\left(t|t-1\right)\geq 0. \end{array}$$From (Equation 72), defining $\Delta P\left(t|t+N\right)=E[\tilde{x}\left(t|t+N\right){\tilde{x}}^{T}\left(t|t+N\right)]$, we have
(80)$$\begin{array}{c} \Delta P\left(t|t+N\right)=Q_{0}\left[\begin{array}{cc} \Delta P_{ai}\left(t|t+N\right) & \Delta P_{x\omega }\left(t|t+N\right)\\ \Delta P_{x\omega }^{T}\left(t|t+N\right) & \Delta P_{wai}\left(t|t+N\right) \end{array}\right]Q_{0}^{T}, \end{array}$$
substituting (78) into (Equation 80), yield
(81)$$\begin{array}{cc} \Delta P(t|t+N)= & Q_{0}\left[\begin{array}{cc} \Psi_{N}\left(t\right)\Delta P_{ai}\left(t|t-1\right)\Psi_{N}^{T}\left(t\right) & \Psi_{N}\left(t\right)\Delta P_{ai}\left(t|t-1\right){\Psi_{N}^{w}}^{T}\left(t\right)\\ \Psi_{N}^{w}\left(t\right)\Delta P_{ai}\left(t|t-1\right)\Psi_{N}^{T}\left(t\right) & \Psi_{N}^{w}\left(t\right)\Delta P_{ai}\left(t|t-1\right){\Psi_{N}^{w}}^{T}\left(t\right) \end{array}\right]Q_{0}^{T}\\ \; & +Q_{0}\left[\begin{array}{cc} \sum_{r=0}^{N}\overset{˘}{K}\Delta \Lambda_{i}\left(t+r\right){\overset{˘}{K}}^{T} & \sum_{r=0}^{N}\overset{˘}{K}\Delta \Lambda_{i}\left(t+r\right){\overset{˘}{M}}^{T}\\ \sum_{r=0}^{N}\overset{˘}{M}\Delta \Lambda_{i}\left(t+r\right){\overset{˘}{K}}^{T} & \sum_{r=0}^{N}\overset{˘}{M}\Delta \Lambda_{i}\left(t+r\right){\overset{˘}{M}}^{T} \end{array}\right]Q_{0}^{T}, \end{array}$$
then (Equation 81) can be rewritten as
$$\begin{array}{cc} \Delta P(t|t+N)= & Q^{\left(0\right)}\left[\begin{array}{cc} \Delta P_{ai}\left(t|t-1\right) & \Delta P_{ai}\left(t|t-1\right)\\ \Delta P_{ai}\left(t|t-1\right) & \Delta P_{ai}\left(t|t-1\right) \end{array}\right]{Q^{\left(0\right)}}^{T}\\ \; & +\sum_{r=0}^{N}Q^{\left(1\right)}\left[\begin{array}{cc} \Delta \Lambda_{i}\left(t+r\right) & \Delta \Lambda_{i}\left(t+r\right)\\ \Delta \Lambda_{i}\left(t+r\right) & \Delta \Lambda_{i}\left(t+r\right) \end{array}\right]{Q^{\left(1\right)}}^{T}, \end{array}$$
where $Q^{\left(0\right)}=Q_{0}\mathrm{diag}\left\{\Psi_{N}\left(t\right),\;\Psi_{N}^{w}\left(t\right)\right\},\overset{˘}{K}=\left[K_{r}^{Nw}\left(t\right),\;K_{r}^{Nv}\left(t\right)\right],\overset{˘}{M}=\left[M_{r}^{Nw}\left(t\right),\;\;M_{r}^{Nv}\left(t\right)\right],Q^{\left(1\right)}=Q_{0}\mathrm{diag}\left\{\left[K_{r}^{Nw}\left(t\right),\;K_{r}^{Nv}\left(t\right)\right],\;\left[M_{r}^{Nw}\left(t\right),\;\;M_{r}^{Nv}\left(t\right)\right]\right\}$, applying Lemma 1, we haveΔP(t|t+N)≥0. Taking the trace of ΔP(t|t+N), we can easily obtain Δtr(P(t|t+N))≥0. The Proof of Theorem 3 is completed. □

## 6. Simulation

Consider the circuits system shown in Figure 1, $u_{e}\left(t\right)$ is control input, $R_{0}$, $L_{0}$, $C_{1}$ and $C_{2}$ are resister, inductor and capacities, respectively. The MSDS model is given as follows
$$\begin{array}{c} \left[\begin{array}{cccc} C_{1} & 0 & 0 & 0\\ 0 & C_{2} & 0 & 0\\ 0 & 0 & -L_{0} & 0\\ 0 & 0 & 0 & 0 \end{array}\right]\frac{dx(t)}{dt}=\left[\begin{array}{cccc} 0 & 0 & 0 & 1\\ 0 & 0 & 1 & 0\\ -1 & 1 & 0 & 0\\ 1 & 0 & R_{0} & R_{0} \end{array}\right]x\left(t\right)+\left[\begin{array}{c} 0\\ 0\\ 0\\ -1 \end{array}\right]w\left(t\right)+\left[\begin{array}{c} 0\\ 0\\ 0\\ -1 \end{array}\right]u\left(t\right) \end{array}$$
where, $x\left(t\right)={\left[u_{e1}\left(t\right),u_{e2}\left(t\right),i_{1}\left(t\right),i_{2}\left(t\right)\right]}^{T}$, $u_{e1}\left(t\right)$ and $u_{e2}\left(t\right)$ are the voltage of $C_{1}$ and $C_{2}$, $i_{1}\left(t\right)$ and $i_{2}\left(t\right)$ are the current of $C_{1}$ and $C_{2}$, w(t) is zero mean white noise, the variance is $Q_{W}$.

Taking the sample period $T_{0}=0.1$ s, the brief parameter matrices are as follows: $$\begin{array}{c} M=\left[\begin{array}{cccc} C_{1} & 0 & 0 & 0\\ 0 & C_{2} & 0 & 0\\ 0 & 0 & -L_{0} & 0\\ 0 & 0 & 0 & 0 \end{array}\right],\Gamma =\left[\begin{array}{c} 0\\ 0\\ 0\\ -1 \end{array}\right],\Phi =M+T_{0}\left[\begin{array}{cccc} 0 & 0 & 0 & 1\\ 0 & 0 & 1 & 0\\ -1 & 1 & 0 & 0\\ 1 & 0 & R_{0} & R_{0} \end{array}\right],B=\left[\begin{array}{c} 0\\ 0\\ 0\\ -T_{0} \end{array}\right]. \end{array}$$

Let $u_{e1}\left(t\right)=0.1$, $C_{1}=2$, $C_{2}=10$, $L_{0}=1$, H=[0,1,0,1], $\lambda_{\alpha }=0.9$, $\lambda_{\beta }=0.9$, $\lambda_{\gamma }=0.9$, $Q_{W}=1.5$, R=4, $P_{0}=10^{2}I_{4}$. Furthermore, the following matrices in (15) as given as
$$\begin{array}{c} M_{1}=\left[\begin{array}{ccc} 10 & 0 & 0\\ 0 & 2 & 0\\ 0 & 0 & 1 \end{array}\right],P=\left[\begin{array}{cccc} 0 & 1 & 0 & 0\\ 1 & 0 & 0 & 0\\ 0 & 0 & 1 & 0\\ 0 & 0 & 0 & 1 \end{array}\right],Q=\left[\begin{array}{cccc} 0 & 1 & 0 & 0\\ 1 & 0 & 0 & 0\\ 0 & 0 & -1 & 0\\ 0 & 0 & 0 & 1 \end{array}\right]. \end{array}$$

Figure 2 and Figure 3 gives the first and second components of actual state $x_{1}$, $x_{2}$ and corresponding filters $x_{1}\left(t|t\right)$, $x_{2}\left(t|t\right)$ from t=600 to t=1200, where the solid curves denote the true state components x(t) and the dotted curves denote $x_{p}\left(t|t\right)$. From Figure 3, the every component of robust filter can effectively follow the true state component $x_{p}\left(t\right)$.

To verify the correctness of the obtained robust Kalman estimator, a Monte Carlo simulation is performed, and the mean square error (MSE) curve of the robust time-varying estimator is shown in Figure 4, Figure 5 and Figure 6. It is easy to see that the value of MSE(t|t+N) can be approximated to the value of trP(t|t+N), and as Theorem 3 states, it has an upper bound trP(t|t+N).

In Figure 4, Figure 5 and Figure 6, the dashed black line shows the trace of the actual estimated error variance, the curved line shows the MSE value, and the dashed orange line shows the actual upper bound on the variance of the estimation error.

**Remark 2.** 
*Time delay is not considered in references [20,21,22,23,24,25,26,27]. Meanwhile, references [19,20,21] do not consider missing measurement, references [19,21,27] ignore the multiplicative noise, and references [19,20,25,27] do not consider packet dropouts. In Table 1, the model of this paper contains more influencing factors, and it is more general than references [19,20,21,22,23,24,25,26,27].*


## 7. Conclusions

In this paper, the robust Kalman estimation of multi-sensor linear singular systems is studied. The singular value decomposition (SVD) method, the augmented state method and the fictitious noise method are applied to transform the original generalized system into a new standard system with uncertain-variance noise. Based on the minimum–maximum robust estimation principle and Kalman filtering theory, a new robust Kalman estimator for augmented systems is obtained. According to the relationship between the augmented state and the original system state, the robust Kalman estimator of the original system is given. Using mathematical induction and the Lyapunov equation method, the robustness of the actual Kalman estimator to the original system is proved. In the future, we will investigate time-varying robust Kalman estimators for a multi-sensor descriptor system with a measurement delay and packet loss. Furthermore, we will consider an uncertain multi-sensor descriptor system in which multiplicative noise occurs simultaneously in both the system and the measurement models, and study the corresponding Kalman filter.

The limitation of this paper is that it uses a general method for studying singular systems. In the future, we will explore some novel methods to study the problem of robust estimation of multi-sensor singular systems

## Figures and Tables

**Figure 1 sensors-23-06968-f001:**
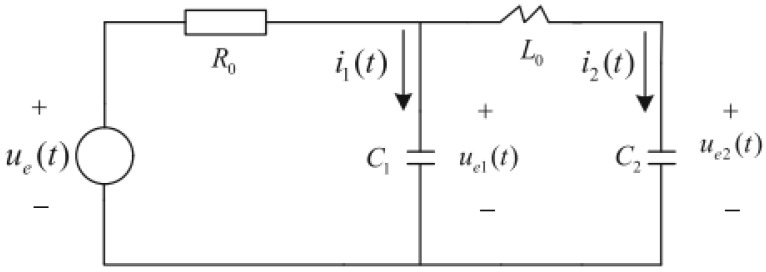
The circuit system.

**Figure 2 sensors-23-06968-f002:**
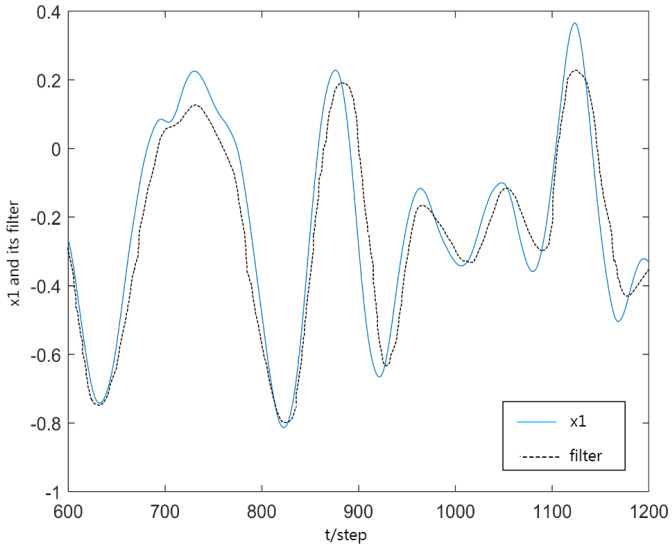
$x_{1}$ and its filter ${\hat{x}}_{1}\left(t|t\right)$.

**Figure 3 sensors-23-06968-f003:**
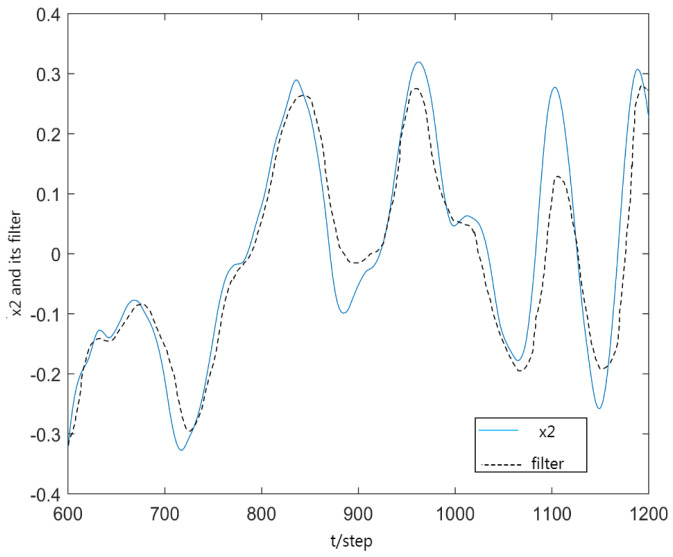
$x_{2}$ and its filter ${\hat{x}}_{2}\left(t|t\right)$.

**Figure 4 sensors-23-06968-f004:**
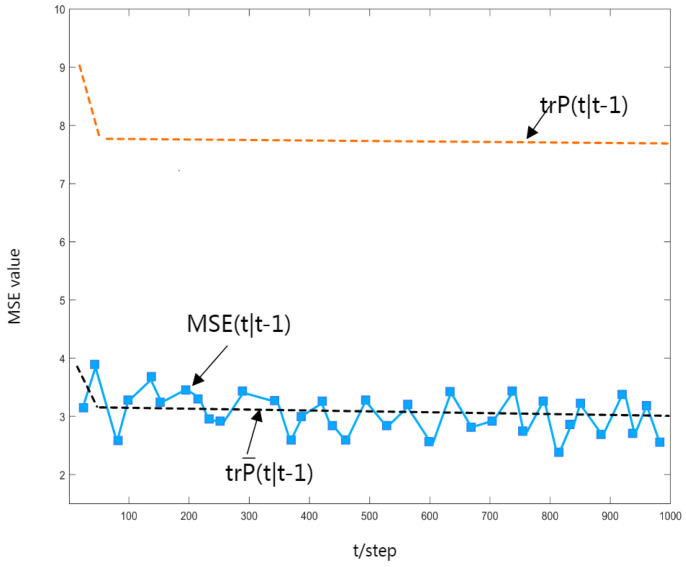
MSE(t|t−1) curve.

**Figure 5 sensors-23-06968-f005:**
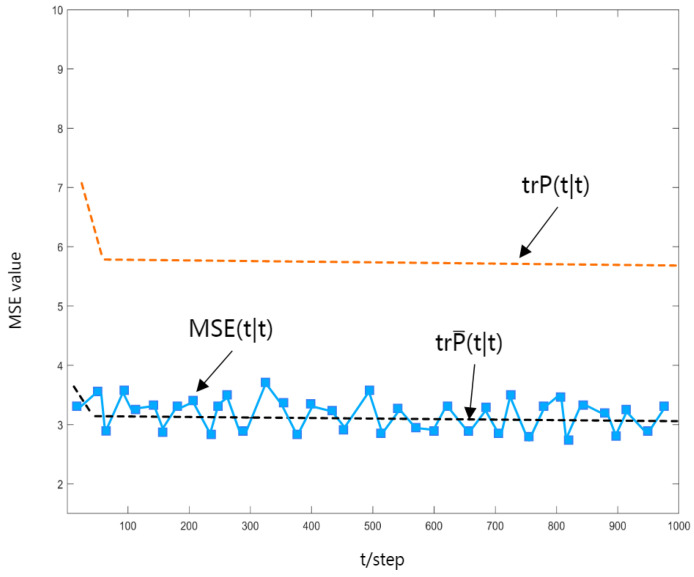
MSE(t|t) curve.

**Figure 6 sensors-23-06968-f006:**
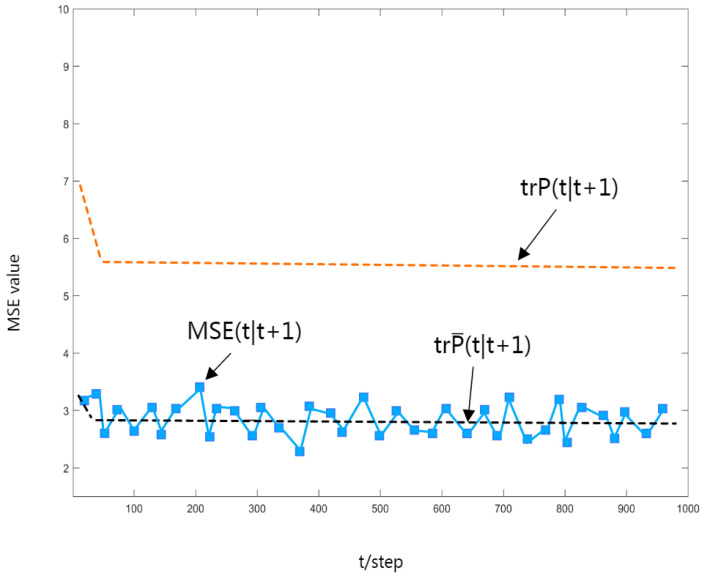
MSE(t|t+1) curve.

**Table 1 sensors-23-06968-t001:** Model comparisons.

Model	This Paper	[19]	[20]	[21]	[25]	[27]
Uncertain-variance noise	√	√	√	√	√	√
Multiplicative noise	√	×	√	×	√	×
Missing measurement	√	×	×	×	√	√
Time delay	√	√	×	×	×	×
Packet dropouts	√	×	×	√	×	×
Multi-sensor descriptor system	√	×	√	√	×	√

Where “√” means that the model contains this component, and “×” means that the model does not contain this component.

## Data Availability

Not applicable.

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
