# Peer review of "Robust Fusion Kalman Estimator of the Multi-Sensor Descriptor System with Multiple Types of Noises and Packet Loss"

_sensors, 2023, doi:10.3390/s23156968_

Round 1
Reviewer 1 Report
The paper addresses the problem of uncertainties and data loss in multi-sensor descriptor system by means of several cascaded approaches: singular value decomposition, augmented state, fictitious noise approach, robust min max estimator, Kalman filter, predictor, smoother and white noise deconvolution estimator, robust fusion Kalman, induction method and Lyapunov equation. The proposed method, as well as almost the entire extensive mathematical discussion, seems very similar to the ones illustrated in
[Zheng, J., Ran, C. Distributed fusion robust estimators for multisensor networked singular control system with uncertain-variance correlated noises and missing measurement. Comp. Appl. Math. 42, 66 (2023)]
The original contribution respect to
[Wang X, Liu W, Deng Z (2017) Robust weighted fusion Kalman estimators for systems with multiplicative noises, missing measurements and uncertain-variance linearly correlated white noises. Aerosp Sci Technol 68:331–344]
should be also highlighted.
Reviewer 2 Report
This paper analyzed the robust Kalman estimation problem studied for the multi-sensor descriptor system. The estimator consists of four parts, which are the usual Kalman filter, predictor, smoother and white noise deconvolution estimator. I have the following concerns:
- Please add information about the remainder of the paper after the main contribution. This information helps readers to follow easily.
- The related works are insufficient to get the current state and gaps of this research field, and AI-based methods in the last five years must also be added as Section 2.
- Please compare the results with the existing method by adding figures and tables.
- Add a new section named “Limitation and Discussion” to give a limitation of the proposed method and future research gaps in this field.
Reviewer 3 Report
The manuscript is devoted to the problem of data processing in the context of the measurement series. The authors consider the descriptor model (1)-(4) and conduct detailed study of its performance. In my opinion, the paper is interesting and can be considered for publication after the issues presented below are addressed
1) The authors have to underline which components of the MSDS model (1)-(4) are new as compared to other studies? The meaning of these expressions also must be described in detail.
2) In Eq. (2): why the authors consider noise per sensor as sum of na noises, not as some resulting single noise? It looks like some unnecessary complication.
3) There is too much of mathematics inside the text paragraphs. The text between Eqs. (21) and (26) is completely unreadable. The same concerns the the text in the page 6.
In the text between Eqs. (24) and (25), the authors should explain why white noise consists of 4 components.
4) Introduction is badly written. The authors must better tell to a reader how descriptor systems considered relate to real-world measurements.
5) There are many typos (for example, "time dalay" in the abstract, "kalman" without a capital K) and linguistic errors. Definitely there is something wrong with the first sentence of the Introduction. The verb "define" is used incorrectly throughout the paper.
Reviewer 4 Report
In this paper, the robust Kalman estimation problem for the multi-sensor descriptor system is studied.
The paper is based on mathematical modelling of Kalman filtration. I couldn't check all the equations, but I assume they are fine. This follows from the experiments.
GPS positioning row 36 - today GNSS, all systems use Kalman filtration, not only GPS
In the introduction, add may be a figure or a scheme (flowchart) of classical Kalman filtaration problem and solving. This will help the reader to better understand the text.
To verify the correctness of the obtained robust Kalman estimator, some experiments were made (row 132).
Figures 1 and 2 show smoothing, but it is not noticeable at first look. Describe it better.
there are many mathematical expressions in the text, but only some are numbered. Does this have any meaning? I understand there are many.
To verify the correctness of the obtained robust Kalman estimator, a Monte Carlo simulation is performed (row 132 -136).
However, it is necessary to explain better for the reader, not everyone will understand this figure description (fig.4 MSE(t|t - 1) curve.).
Thus it cannot be said for the uninformed that your theorem has been successfully verified.
The conclusion is very short, should be extended and focused on benefits of your reserch and comparison with an independent verification.
Also add to the conclusion where your approach will be used to advantage and what it will deliver.
Also, there should be more citations, especially outside of Asia; you are using Kalman filtering, so there should be a citation on this topic.
See for example:
https://en.wikipedia.org/wiki/Kalman_filter
Kalman, R.E. (1960). "A new approach to linear filtering and prediction problems" (PDF). Journal of Basic Engineering. 82 (1): 35–45. doi:10.1115/1.3662552. Archived from the original (PDF) on 2008-05-29. Retrieved 2008-05-03.Kalman, R.E.; Bucy, R.S. (1961). "New Results in Linear Filtering and Prediction Theory". Journal of Basic Engineering. 83: 95–108. CiteSeerX 10.1.1.361.6851. doi:10.1115/1.3658902.
Round 2
Reviewer 3 Report
The authors did not succeed in improving the manuscript. There are still very long mathematical expressions inside the text that significantly worsens readability. The Introduction is still unclearly written. The authors have to consistently state first the general statement of the problem in a way that would be understandable to the reader. Then they can proceed with particular statement of the problem. The paper is still not ready for publication.
Reviewer 4 Report
Dear authors, ok, thank you for improving the article and I have no further questions
Author Response
Thank you for your valuable time and positive comments.